# An integrative method to decode regulatory logics in gene transcription

Bin Yan[1], Daogang Guan[2,3], Chao Wang[2,3], Junwen Wang[4], Bing He[2,3], Jing Qin[5], Kenneth R. Boheler[1,6], Aiping Lu[2,3], Ge Zhang[2,3] & Hailong Zhu[2,3]

Modeling of transcriptional regulatory networks (TRNs) has been increasingly used to dissect the nature of gene regulation. Inference of regulatory relationships among transcription factors (TFs) and genes, especially among multiple TFs, is still challenging. In this study, we introduced an integrative method, LogicTRN, to decode TF–TF interactions that form TF logics in regulating target genes. By combining *cis*-regulatory logics and transcriptional kinetics into one single model framework, LogicTRN can naturally integrate dynamic gene expression data and TF-DNA-binding signals in order to identify the TF logics and to reconstruct the underlying TRNs. We evaluated the newly developed methodology using simulation, comparison and application studies, and the results not only show their consistence with existing knowledge, but also demonstrate its ability to accurately reconstruct TRNs in biological complex systems.

[1] Centre of Genomics Sciences, LKS Faculty of Medicine and School of Biomedical Science, University of Hong Kong, Hong Kong, China. [2] School of Chinese Medicine, Hong Kong Baptist University, Hong Kong, China. [3] Institute of Integrated Bioinformedicine and Translational Science, Hong Kong Baptist University, Hong Kong, China. [4] Department of Health Sciences Research and Center for Individualized Medicine, Mayo Clinic, & Department of Biomedical Informatics, Arizona State University, Scottsdale, AZ 85259, USA. [5] School of Life Sciences, The Chinese University of Hong Kong, Hong Kong, China. [6] Division of Cardiology, Johns Hopkins University School of Medicine, Baltimore, MD 21205, USA. Bin Yan and Daogang Guan contributed equally to this work. Kenneth R. Boheler, Aiping Lu, Ge Zhang and Hailong Zhu jointly supervised this work. Correspondence and requests for materials should be addressed to A.L. (email: aipinglu@hkbu.edu.hk) or to G.Z. (email: zhangge@hkbu.edu.hk) or to H.Z. (email: hlzhu@hkbu.edu.hk)

Unraveling mechanisms of transcriptional gene regulation in eukaryote organisms is fundamental to the understanding of biological processes responsible for development of organs, progression of diseases and other complex biological events like aging. To unravel these mechanisms, systems biology approaches aim to model regulatory relationships among molecules in a biological system as a whole rather than as individual entities. A robust systems biology approach is possible only through application of computational modeling theory with high-throughput technologies. Modeling approaches are mainly devoted to infer transcriptional regulatory networks (TRNs) that accommodate direct regulations from TFs to their target genes. TRNs can be inferred using either gene expression data or DNA-binding information[1, 2]. Derived networks based exclusively on expression data represent possible functional relationships among genes, but not all relationships are associated with DNA binding and altered gene function. High-throughput TF-DNA-binding data provides abundant information to investigate the physical interactions between TFs and genes. Coupling chromatin immunoprecipitation with next-generation sequencing (ChIP-seq) allows high fidelity mapping of TFs to genomic locations[3], and detection of TF-binding regions on DNA sequences. Thus, TF-DNA-binding signals can be employed to reconstruct TRNs. However, TF binding is not always functionally involved in the regulation of gene transcription. The complexity of these approaches thus poses significant challenges to computational biologists.

Integrative analyses that interrogate different types of omics data are more appealing in revealing transcriptional relationships than non-integrative analyses. A number of studies have made useful attempts to predict TF target genes by combining gene expression profile and TF-binding data. Examples include NCA[4] and fastNCA[5] using matrix decomposition, APG using Graphical Gaussian model[6], linear model with LASSO GEMULA[7], and COGRIM using Bayesian hierarchical algorithm[8]. The Dialogue for Reverse Engineering Assessments and Methods (DREAM)

started a concerted effort by developing various algorithms to understand the transcriptional gene regulation from high-throughput data[9-11]. Previously, we developed integrative methodologies to identify network-based TF target genes, such as sparse matrix decomposition[12], and Graphical Gaussian model with partial least squares[13]. Although most of these methods show a capacity to integrate the two types of data, it remains challenging to capture co-regulatory features of TFs. This is because the inferred TRNs characterize neither the interactive nature between TFs in regulating the target genes nor the transcriptional dynamics. Consequently, these existing methods or approaches cannot effectively decode the cooperative regulation of multiple TFs on dynamic gene expression. Thus, the resultant networks are still far from being adequate to model true transcriptional gene regulation.

Logic gates, which utilize more than one input, have long been used to describe complex interactive relationships among TFs[14]. The regulatory function specified in *cis*-regulatory sequence in sea urchin system was represented as Boolean logics among seven TF-binding sites that direct the spatial expression and repression of the Endo16 gene[15]. In vertebrate neural tube, the regulatory logic of a transcriptional network which links three TFs to Sonic Hedgehog signaling, was found to be responsible for differential spatial and temporal gene expression[16]. In the skeleton-genic mesoderm of sea urchin embryo, a regulatory logic of double negative gate was able to prevent the expression of a global transcriptional repressor that establishes the skeleton-genic regulatory state through a lineage-specific repressor, Pmar-1[17]. Typically, regulatory logics were applied on binary variables, which only provide qualitative features, and thus may not be adequate to describe the complicated dynamic behaviors in gene expression[18-20].

In this study, we present a comprehensive methodology, LogicTRN, by integrating gene expression data and TF-DNA-binding information to decipher TF regulatory logics in gene transcription. The newly developed method can quantitatively characterize logic relations between TFs by combining

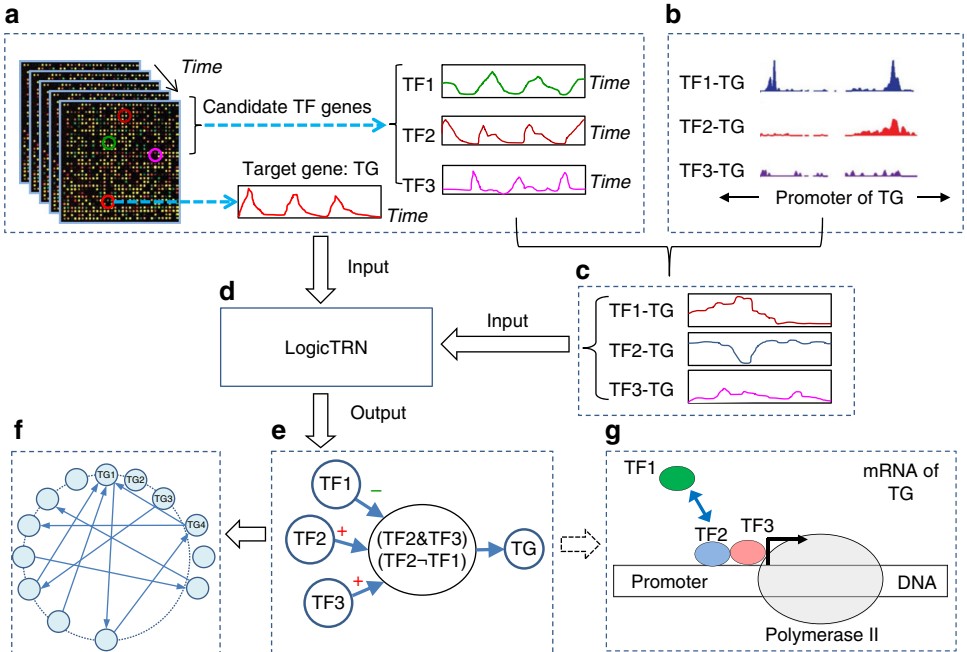

**Fig. 1** The computational framework of LogicTRN. **a** Time-series gene expression data. For a given target gene G1, we assume three TFs, TF1, TF2, and TF3, can bind to its promoter. **b** TF-DNA-binding signals from ChIP-seq measurement or TF-binding motifs on target gene promoters. **c** Calculation of TF-DNA-binding occupancy over time. **d** Construction of model equations using the data from **a**, **c**. **e** Identification of the regulatory logic for the gene TG. **f** Reconstruction of the TRN consisting of TF regulatory logics and target genes. **g** Describing transcriptional gene regulatory mechanism

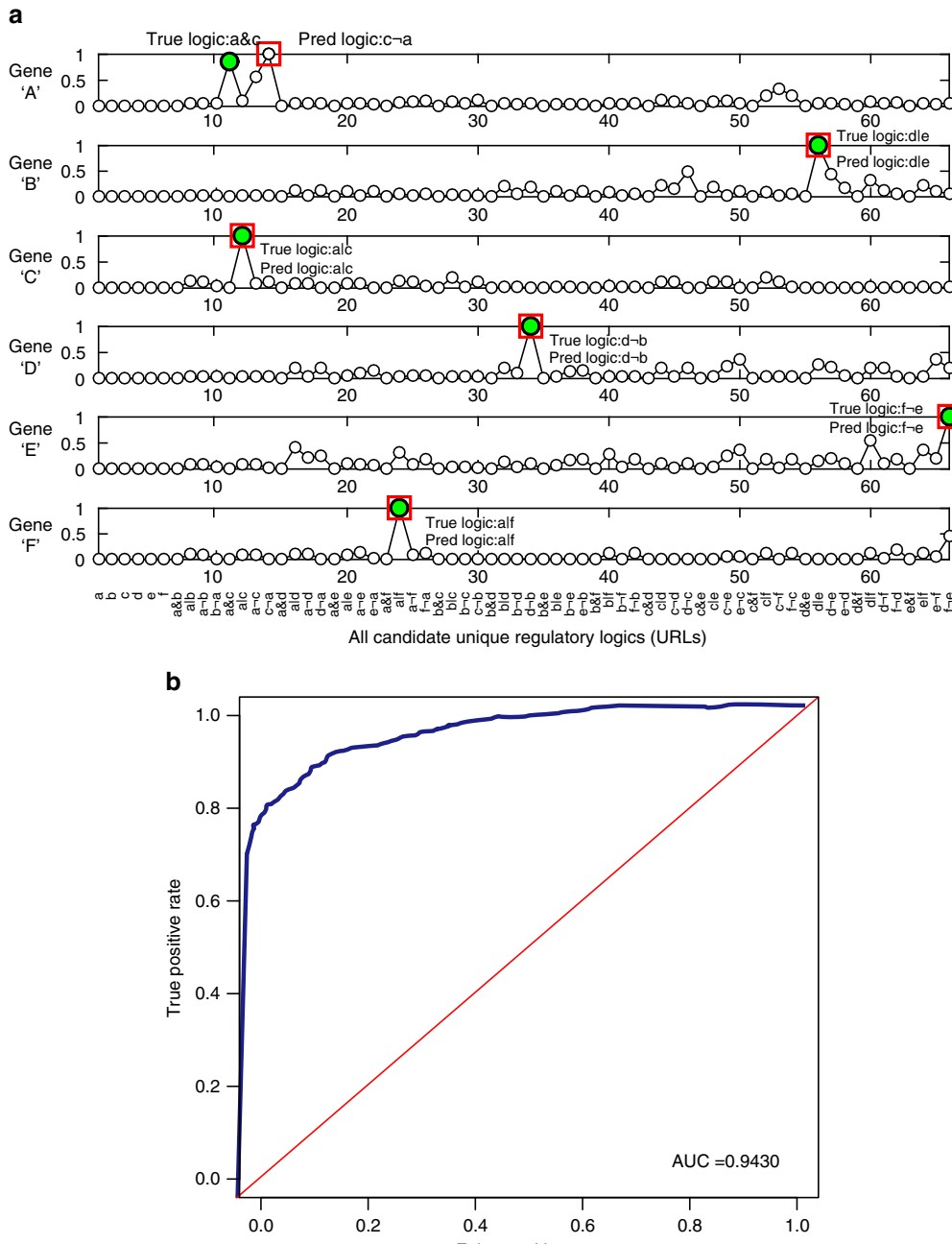

**Fig. 2** Identification of URLs in a simulation study. **a** Each subplot corresponds to a target gene, in which the *x*-axis represents all 66 URLs, and the *y*-axis represents the predicted probabilities of the URLs (normalized by dividing with the largest value), as illustrated in small white circles '○'. The true regulatory logics are marked with green circles, while the predicted logics are marked with red squares. It can be seen that LogicTRN correctly identifies the regulatory logics for five of six genes. **b** ROC curve and AUC of network reconstruction in the simulation study. The average ROC curve and its dynamic range are obtained from 1000 runs with random settings in each run. The red line represents the performance of random guesses

*cis*-regulatory logics and transcriptional kinetics in one single model framework. The derived TF logics are then used to infer the putative TF cooperation in regulating target genes so as to reconstruct TRNs. We conducted simulation and comparison studies to evaluate the performance of LogicTRN. Then we applied this method to analyze data sets representing the estrogen-induced breast cancer and human-induced pluripotent stem cell (hiPSC)-derived cardiomyocyte (CM) development. The derived networks are able to explore the nature of transcriptional gene regulation with biological meanings, which are consistent with previously experiments. Successful application of LogicTRN shows its ability to identify TF targets and to reconstruct TRNs.

## Results

**Computational framework of LogicTRN**. The computational framework of LogicTRN is illustrated in Fig. 1. Briefly, to identify TF regulatory logics in a biological process, two types of data are required to start LogicTRN, including time series gene expression profiles (A) and TF-DNA-binding signals (B). For a given gene TG, TF1, TF2, and TF3 are assumed as three putative TFs that can bind its promoter. We extract the dynamic expression data of TG from (A), and the binding signals of the three TFs on TG from (B). Based on (A) and (B), the TF-DNA-binding occupancies can then be estimated (C). Third, the transcriptional regulatory model in LogicTRN is represented with a group of

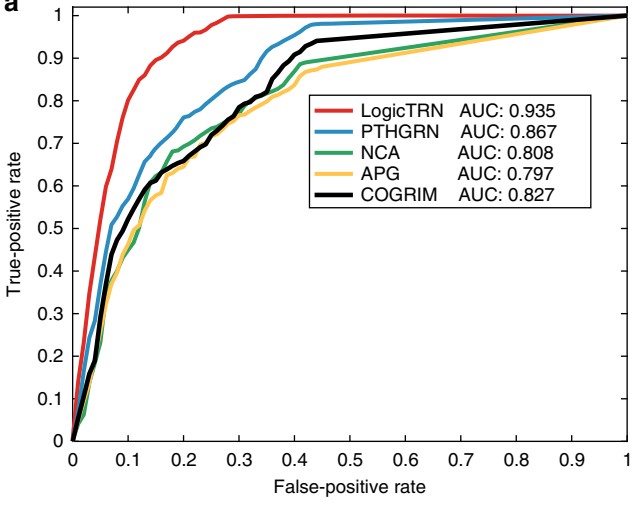

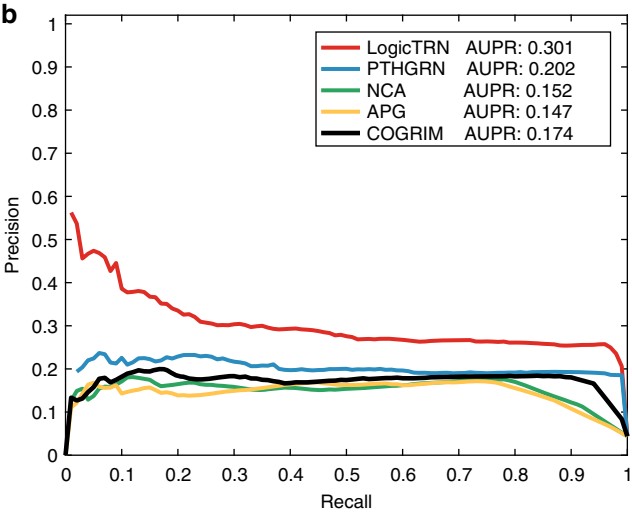

**Fig. 3** Comparison of LogicTRN with COGRIM, APG, NCA, and PTHGRN in identifying target genes of mouse ESC TFs (Oct4/Pou5fl, Sox2, Nanog, and Suz12). **a** ROC curves and their AUCs. **b** PR curves and their AUCs. To conduct ROC and PR analyses, a full set of positives and negatives were employed (see Supplementary Data 3)

model equations with the inputs of gene expression data and TF-DNA-binding occupancies. Solving the model equations can lead to identification of the regulatory logics of the target gene TG (D) using the method described in Methods section and Supplementary Information. Two resultant logics, TF2&TF3, and TF2¬TF1 are identified to regulate TG (E). Repeatedly applying the same procedure for each gene, we thus can reconstruct a TR`N that links all the TF logics and their regulated genes (F). The identified logics of a target gene provide a hypothesis for exploring the underlying mechanism of its transcriptional regulation (G).

**Simulation study**. To evaluate the reliability of LogicTRN, we constructed a simulation platform of six nodes representing TF proteins (when used as regulators) and genes (when used as targets). The platform was configured to simulate the process of transcriptional gene regulation to generate dynamic gene expression data and TF-DNA-binding occupancy after setting the regulatory logics and kinetic parameters. The kinetics of TF gene expression on each node are described as $dy_m(t)/dt = I_s(t) - k_{dm}y_m(t)$, where $y_m(t)$ is gene expression at time $t$, $I_s(t)$ is the gene initiation rate, and $k_{dm}$ is gene degradation

rate. The kinetics of TF protein are defined as $dP(t)/dt = K_{Tr}y_m(t) - k_P y_m(t)$, where $y_m(t)$ is gene expression at time $t$, $I_s(t)$ is the gene initiation rate, and $k_{dm}$ is gene degradation rate. The kinetics of TF protein are defined as $dP(t)/dt = K_{Tr}y_m I_s(t) - k_P y_m(t)$, where $K_{Tr}$ is protein translation rate, $P(t)$ is protein concentration level and $k_P$ is protein degradation rate. The TF-DNA occupancy is defined by $Y(t) = (K_r P(t))/(D_n + K_r P(t))$, where $K_r$ is the relative equilibrium constants defined as the ratio between the specific and the non-specific equilibrium constants, which normally ranges from $10^4$ to $10^6$. $D_n$ represents the number of unoccupied nonspecific sites, which can be regarded as a constant approximated using 90% of the total genome.

For simplification purposes, we restricted each gene to regulation by at most two TFs (self-regulation is allowed). Thus, a total set of 66 combinatorial logics is available for each gene. The ranges of kinetic parameters are set as follow: messenger RNA (mRNA) degradation rate ranges from 0.005 to 0.015 (percentage per minute), the regulatory strength ranges from 1.0 to 3.0; the maximal transcription initiation rate is from 20 to 100 mRNA per minute. Other parameters are fixed as below: gene transcriptional delay is set to 20 min; protein translational delay 20 min; protein translational rate 2 molecules/mRNA*minute; protein degradation rate is set to 0.01, 0.009, 0.008, 0.007, 0.006, and 0.005 percentage per minute for protein a, b, c, d, e, and f, respectively; and relative equilibrium constants on TF Gene A, B, C, D, E, and F were fixed as 10,000, 14,000, 20,000, 24,000, 30,000, and 40,000, respectively.

Based on the data generated by the platform, LogicTRN was implemented to predict the TF logic on each gene. Implementation of LogicTRN is as follows: Step one, use all of the six TFs as potential regulators for each gene. Step two, construct the signatures of 66 unique regulatory logics (URLs) based on all 72 composite variables (Supplementary Data 1). Step three, construct the model equations using the gene profiles and TF-DNA-binding occupancy data generated by the simulation platform. Step four, conduct LASSO regression[21, 22] on the group of model equations of a gene to obtain a coefficient matrix. Step five, calculate the confidence value of each URL according to the coefficient matrix. The URL with the highest confidence value will be chosen as the dominant logic. As such, both the regulator TFs and logic can be determined for a target gene.

Figure 2 shows the results of the simulation. In Fig. 2a, each subplot corresponds to a target gene, in which x-axis shows all the 66 URLs and y-axis represents the confidence value. The confidence values of all the URLs are illustrated with small white circles. The true regulatory logics are marked with solid green circles, while the predicted logics are marked with red squares. It can be seen that LogicTRN correctly identified the regulatory logics for five out of six genes in this example. To evaluate the overall accuracy and reliability of LogicTRN in logic prediction, we performed the above procedure 1000 times. For each run, the regulatory logics and kinetic parameters were randomly set for each gene. We used the receiver operating characteristic curve (ROC) to evaluate the accuracy of logic identification. Upon the ranked logics of a gene predicted by LogicTRN, the threshold of confidence value was gradually reduced to involve more logics in prediction. The true-positive rate (TPR) and false-positive rate (FPR) of logic identification were obtained by comparing the prediction with the true logics. The ROC is shown in Fig. 2b, and the area under the curve (AUC) is 0.943, suggesting that logics generated by LogicTRN is highly accurate and reliable.

**Comparison study**. To further examine the efficiency or accuracy of LogicTRN in identifying the target genes of TFs, we compared

LogicTRN with existing methods COGRIM[8], APG[6], NCA[4], and PTHGRN[13] that are capable of integrating gene expression and TF-DNA-binding data. The four methods and LogicTRN were independently applied to run mouse embryonic stem cell (mESC) data set (Supplementary Data 2, Supplementary Note 1). Since these existing approaches do not accommodate TF regulatory logics, we only compared the methods on the accuracy of predicting TF target genes. Figures 3a, b show the ROC and Precision-Recall (PR) curves of the five methods on mESC, respectively. The area under the ROC curve and PR curves, namely ROC-AUC and PR-AUC, were calculated to represent the performance of a prediction. It can be seen that LogicTRN has the highest ROC-AUC (0.935) among the five methods on the mESC data set, which are much higher than the ROC-AUCs of other methods. Similarly, the PR-AUC of LogicTRN is 0.301 (using all negatives), again is the highest among all the methods. Both results indicate that LogicTRN is a more accurate method for identification of TF target genes.

**TRNs in the E2-induced breast cancer development.** The deregulation of Estrogen Receptor alpha (ESR1) is a major factor causing pathogenesis of breast cancer. Estrogen Receptor-positive tumors, accounting for 60–70% of breast cancer, use steroid hormone estradiol (E2) as their main growth stimulus. ESR1 is able to interact with many TFs, cofactors, and growth factor-activated membrane pathways to form the regulatory machinery which will modulate cancer-related biological processes[23, 24]. To evaluate the applicability of LogicTRN in biological systems, we identified regulatory logics formed by ten breast cancer-related TFs ESR1, FOXA1, FOXM1, GATA3, CEBPB, JUN, FOS, JUND, EP300, and CTCF on regulating target genes, by integrating times series gene expression data of the E2-induced tumor progression and ChIP-seq-binding data of these TFs (Supplementary Note 2).

We first determined how the ten TF genes are regulated by logics. Figure 4 shows logic networks linking the ten TFs at early stages (T1–T2, Fig. 4a) and late stages (T2–T3, Fig. 4b), in which the three parts at left, middle, and right represent regulators, logics and target genes, respectively. The resultant networks indicate that 15 (early stages, Fig. 4a) and 10 (late stages, Fig. 4b) logics control the dynamic transcription of the TF genes. Noticeably, ESR1 and GATA3 logics are dominant, and in particular at the early stage. ESR1 is well known as a master player by interacting with other TFs, that were previously implicated in breast cancer, such as GATA3, FOXA1, and EP300[25], AP1 (JUN and FOS)[26, 27], FOXM1[28], and CEBPB[29]. In an inferred TRN, ESR1 interacts with other TFs such as AP1, GATA3, and CEBP to mediate distinct biological functions in breast cancer cells[30]. Our analysis validates ESR1 function as a main regulator linking cancer-related TFs, especially GATA3 to promote breast cancer development.

Next, we investigated transcriptional regulation of TF logics on their downstream genes. Supplementary Data 4 provides the list of target genes of all the identified logics, in which top ranked logics were found to regulate a majority of the differential genes at the two stages. It is noticed that these differential target genes of the top ranked logics recruited by LogicTRN mainly involve cell cycle, cell proliferation, and apoptosis (Table 1). We evaluated the enrichment of the predicted logics and their target genes among all the differentially expressed genes by a statistical test (Supplementary Methods). To verify the reliability of the enrichment, we set-up the negative controls (Supplementary Methods) by collecting the logics beyond the enriched logics but have the same TFs involved. As shown in Supplementary Fig. 1A, B, most of the negative controls have higher p-values than the enriched logics in human breast cancer, indicating that top

ranked logics determined by our method can recruit more differentially expressed genes. GATA3 logics are highly enriched and can target the largest number of differential genes at early stages. We re-organized the top 15-ranked logics and their differential targets, and constructed the E2-responsible TRNs (Supplementary Fig. 2). In particular, GATA3 and ESR1 regulate cell cycle and apoptosis by cooperating with other TFs, as shown in Fig. 5. At early stages, GATA3 or ESR1 logics mainly upregulate CDC25A, E2F1, and E2F2 that control G1 to S transition and cell proliferation, and downregulate apoptotic genes BCL2L1 and CXCR4 (Fig. 5a). By contrast, at late stages, ESR1 and GATA3 interact with FOXM1, JUND, and CEBPB to upregulate cyclins B and A, CDC25C and CDC20 that control G2 to M transition (Fig. 5b). The regulation of gene expression at late stages is carried out by logics of ESR1, GATA3, and a broad set of TFs. The LogicTRN analysis provides evidence that GATA3 and ESR1 are key regulators in E2-induced cell proliferation of breast cancer cells. Their regulatory function is by targeting G1 to S and G2 to M of cell cycle at different time periods, respectively.

GATA3 and FOXA1 have been demonstrated as two co-TFs functionally linking with ESR1 in modulating the estrogen response at the transcriptional level[25]. GATA3 was found to act upstream of FOXA1 in mediating ESR1 binding by analyzing ChIP-seq-binding signals in breast cancer cells[31]. In fact, our result supports co-regulation of ESR1 and GATA3 on FOXA1 at the early stage (Supplementary Fig. 7A). In breast cancer, the direct binding of ESR1 on FOXM1 promoter was confirmed both in vitro and in vivo[32]. We repeated this result, and detected joint regulation of ESR1 with CEBPB on FOXM1 during the E2-induced response (Supplementary Fig. 7A). In addition, several other logics are consistent with experimental validation, such as GATA3¬FOXM1[33], GATAs¬EP300[34], and ESR1&GATA3[35]. Our result supports the perspectives regarding role of ESR1 in leading to transcriptional programs underlying breast tumorigenesis[24, 36, 37].

To further validate the predicted logics, we analyzed the influence of various logics on their target genes after TF knockdown (See Methods). According to our model, both TFs in a AND logic, or the activator TF in a NOT logic, should be more crucial to the expression of the target gene. For instance, after knocking down CEBPB, the distribution of GATA3&CEBPB target genes displays significant larger variations comparing to GATA3|CEBPB (p-value = 0.0002, Supplementary Fig. 3A). This result is consistent with the definition of the regulatory logic, suggesting the predicted GATA3&CEBPB is likely to be true. Consistently, the target genes of the predicted CEBPB¬FOXM1 are more affected by knockdown of CEBPB, comparing to FOXM1¬CEBPB (Supplementary Fig. 3A). Meanwhile, the target genes of the predicted ESR1&GATA3 are more affected by knockdown of ESR1, comparing to GATA3¬ESR1 and ESR1|GATA3 (p-value < 0.001, Supplementary Fig. 3B). Similar results were observed when knocking down GATA3 (Supplementary Fig. 3C). The identified E2-responsible TRNs provide evidence that ESR1 form modulators with co-TFs or cofactors to regulate the downstream genes, especially with GATA3 to promote E2-inducd gene expression programs in breast cancer.

**TRNs in hiPSC-derived cardiomyocyte differentiation.** Development of embryonic CM is associated with dynamic changes in gene expression conferring biological functions to CM formation. The altered gene expression could be underlined by a set of cardiac TFs that form TRNs, such as MESP1, MEF2C, GATA4/6, NKX2-5, TBX5, and HAND1/2. These TFs play critical roles in

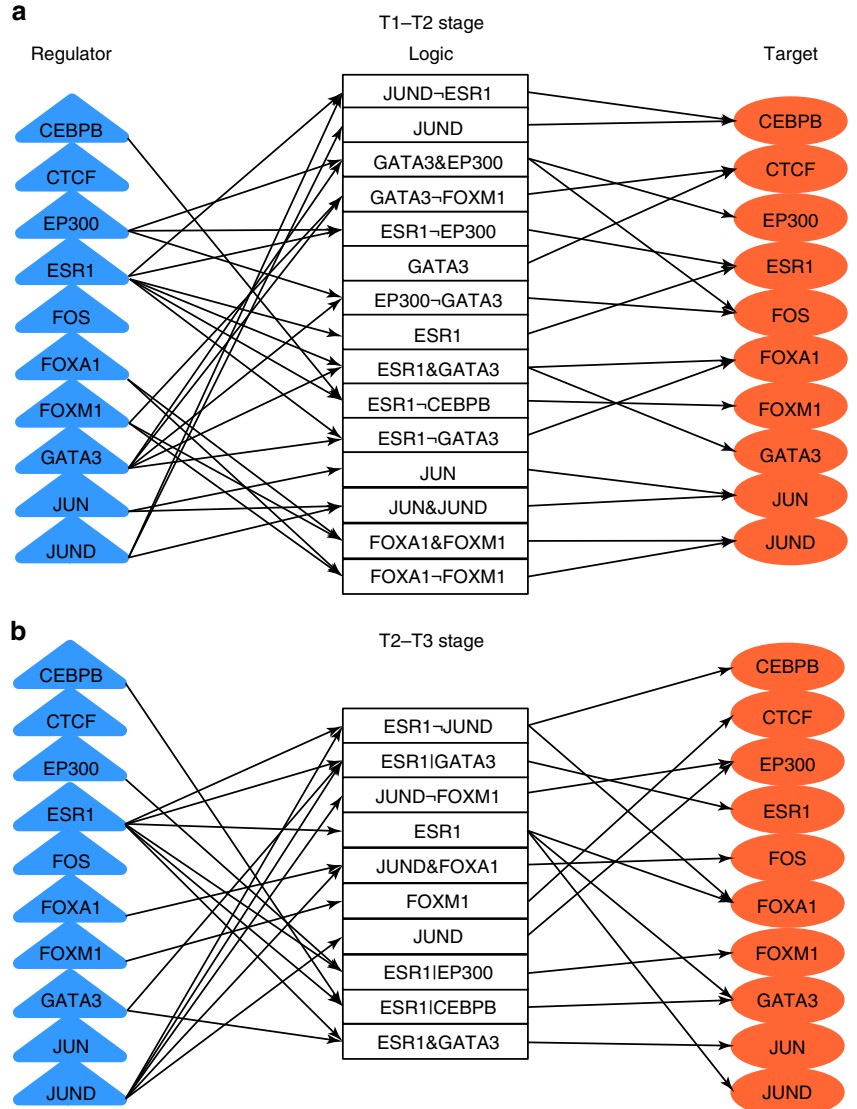

**Fig. 4** Regulatory logics formed by ten TFs during the E2-treated breast cancer development. Time-course gene expression microarray data was employed. The data represents 12 time points 0, 1, 2, 4, 6, 8, 12, 16, 20, 24, 28, and 32 h (h) following the E2 treatment, which was divided into three stages, T1 (0–2 h), T2 (4–20 h), and T3 (24–32 h). **a** represents T1-T2 stages and **b** represents T2-T3 stages. Logic relationships include AND (&), NOT (¬), and OR (|)

regulating cardiac lineage commitment[38–40]. However, it remains unclear how these TFs interact with each other to modulate dynamic progression from human pluripotent stem cells to CM maturation. To solve this challenge, we applied LogicTRN to integrate a time series gene expression profile of the hiPSC-derived CM differentiation with binding data of the eight TFs (Supplementary Note 3).

First, we investigated how the cardiac TFs form logics to regulate themselves. There are 12 logics found to target the eight TF genes at the T1-T2 and T2-T3 stages, respectively (Supplementary Fig. 4). The derived networks show that MESP1 is a primary regulator controlling/triggering transcription of other TF genes, supporting MESP1 function as a master player in determining cardiac cell lineage commitment[41, 42]. This result is consistent with ChIP-seq experiments in mouse showing that Mesp1 directly binds to regulatory DNA sequences located in the promoter of many key cardiac TFs, resulting in a rapid upregulation of *Hand2, Nkx2-5, Gata4, Mef2c*[43]. By contrast, GATA6 interactions with other TFs are likely mediated through different mechanisms, such as competitive or inhibitory actions

**Table 1 Top-ranked TF regulatory logics identified in breast cancer**

| Regulatory logics | No. of differential target genes | Pathways or processes involved |
|---|---|---|
| T1-T2 stage | | |
| GATA3 | 68 | Apoptosis, cell cycle, proliferation |
| CEBPB | 24 | Migration, apoptosis |
| ESR1 | 22 | Cell cycle, proliferation |
| GATA3|CEBPB | 22 | Cell cycle, apoptosis |
| GATA3¬EP300 | 20 | Cell cycle |
| EP300 | 19 | Cell cycle, proliferation |
| T2-T3 stage | | |
| ESR1 | 36 | Cell cycle |
| GATA3 | 27 | Cell cycle |
| FOXM1 | 22 | Cell cycle, proliferation |
| CEBPB | 13 | Migration, apoptosis |
| GATA3|FOXM1 | 11 | Cell cycle, proliferation |
| JUND¬FOXM1 | 10 | Cell cycle |

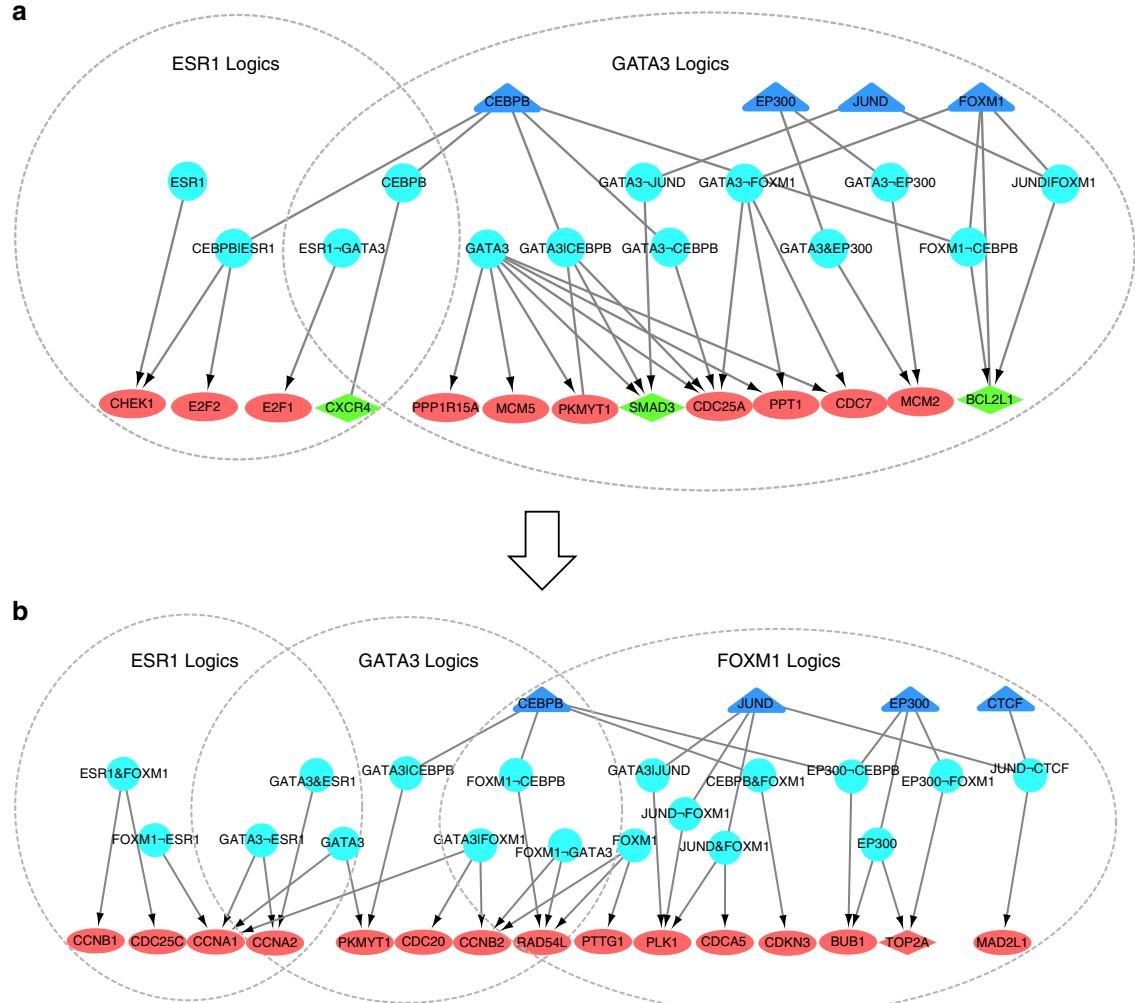

**Fig. 5** E2-responsible TRNs in breast cancer. Time-course gene expression microarray data was employed. The data represents 12 time points 0, 1, 2, 4, 6, 8, 12, 16, 20, 24, 28, and 32 h (h) following the E2 treatment, which was divided into three stages, T1 (0–2 h), T2 (4–20 h), and T3 (24–32 h). For **a** (at T1-T2 stage) and **b** (at T2-T3 stage), triangle nodes in blue refer to TFs, and triangle nodes in light blue are TF logics. The target genes in different pathways of logics are represented in oval nodes (cell cycle: G1 to S at T1-T2 stage, G2 to M at T2-T3 stage) and diamond nodes (apoptosis). Nodes in red and green represent upregulated and downregulated genes, respectively. Logic relationships include AND (&), NOT (¬), and OR (l)

with other TFs such as MESP1, MEF2C, and TBX5 (Supplementary Fig. 4).

Next, we conducted LogicTRN analysis to identify cardiac TF logics in directing CM differentiation. Among all the predicted logics during the three stages of the CM differentiation (Supplementary Data 5), the top ranked logics account for regulating the majority of differential genes. Similarly, we showed the top ranked enriched logics in hiPSC-CM more reliable than negative controls (Supplementary Fig. 1C, D). These differential target genes are mainly involved in heart development, contraction and embryonic development (Table 2). We constructed the cardiac TRNs based on the top 15-ranked logics and their differential downstream genes, and show MESP1 and GATA6 as two dominant TFs to form logics in gene networks associated with heart development, embryonic development, and cell cycle (Supplementary Fig. 5). In particular, MESP1 logics dominate downregulation of pluripotentcy TFs, *SOX2, NANOG, OCT4,* and *ZIC2* that trigger the hiPSC differentiation at early stages (Fig. 6a). Meanwhile, MESP1 is also shown to cooperate with GATA6, as well as other cardiac TFs such as TBX5, HAND1/2, and NKX2-5 to activate expression of genes important for Ca handling (*CACNA1C, SLN, RYR2,* and *ATP2A2*), contraction (*MYL4,*

*MYL7, MYH6,* and *TNNT2*) and heart development (*IRX4, NPPA,* and *NPPB*). These results confirm the function of MESP1 in initiating cardiac lineage differentiation through targeting key components of cardiac relevant pathways or biological processes. At late (T2-T3) stages, MESP1, GATA6, and NKX2-5 logics become more crucial by jointly upregulating the cardiac gene programs (Fig. 6b).

Our prediction is consistent with the published experimental data: MESP1 promotes the expression of cardiac genes *Myh6, Myl2, Myl7,* and *Tnnt2*[42]. Cooperation of GATA4/6 and NKX2-5 is able to facilitate the expression of contractile genes[44, 45]. GATA6 and other GATA members including GATA4 were identified as essential regulators of *NPPA* and *NPPB*[46].

Moreover, we validated the predicted logics by examining the influence of TF overexpression. For example, the target genes of predicted logics MESP1, MESP1¬GATA6, and MESP1&HAND1 are more affected by overexpression of MESP1, compared to other MESP1 logics (Supplementary Fig. 6). Besides, our previous study established a coexpression network among GATA4, HAND1, NKX2-5 and TBX5 associated with human ESCs-derived ventricular development[47]. Supplementary Fig. 7B presents a model showing how the logics formed by MESP1 and

**Table 2 Top-ranked TF regulatory logics in hiPSC-derived CM differentiation**

| Regulatory logics | No. of differential target genes | Pathways or processes involved |
|---|---|---|
| T1-T2 stage | | |
| GATA6 | 103 | Heart, contraction, embryonic |
| MESP1 | 96 | Heart, cell cycle, embryonic |
| MESP1¬GATA6 | 82 | Heart, contraction, embryonic |
| TBX5IMESP1 | 60 | Heart, cell cycle, embryonic |
| GATA6IMESP1 | 57 | Heart, contraction, cell cycle |
| HAND2IMESP1 | 48 | Cell cycle, embryonic |
| T2-T3 stage | | |
| GATA6 | 83 | Heart, contraction, MAPK |
| MESP1&NKX2-5 | 45 | Contraction, cell cycle |
| MESP1 | 37 | Contraction, cell cycle |
| HAND2&MESP1 | 32 | Heart, contraction |
| HAND1¬GATA6 | 30 | Heart, contraction |
| MESP1¬NKX2-5 | 28 | Contraction |

GATA4/6, HAND2, TBX5, and NKX2-5 control transcription of two CM marker genes *TNNT2* and *MYL7*.

Indeed, the LogicTRN-defined logics and the findings support the previous conclusion that MESP1 functions in driving cardiovascular specification[41]. MESP1&GATA6 cooperative logic likely acts as an important cardiac TF module, which would be valuable for further experimental testing. This application reveals logic relations of cardiac TFs and orchestrates dynamic TRNs that instruct sequential differentiation and maturation of hiPSC-derived CM.

## Discussion

A major goal in systems biology is to develop appropriate computational models that are able to integrate various experimental data for dissecting biological complexity in transcriptional gene regulation. To address this challenge, we combine *cis*-regulatory logics and transcriptional kinetics to develop a biologically plausible model. Theoretically, TFs can form many logics in gene regulation, but in reality, only a few of them become the driving force in a biological process. LogicTRN has the ability to identify the dominant logics in an accurate and high-throughput way. In LogicTRN, the change of gene expression is defined as a function of TF-binding occupancies and kinetic parameters. Unlike existing integrative analyses that treat gene expression and TF-DNA binding as two separate processes, our approach puts these two types of data into one single model, which is more biologically meaningful.

One obstacle is that transcriptional kinetic function is essentially nonlinear, which makes it difficult to develop computational methodologies. Here, we adopted Taylor expansion to convert the nonlinear kinetic function to a polynomial function. The usage of the polynomial function provides a general mathematic form to simultaneously involve different regulatory logics. By assuming each logic has a probability of being involved in regulating a target gene, we are able to construct the model equation. Solving the model equations can lead to a determination of the TF logics.

One challenge in developing LogicTRN is that the number of combinatorial logics increases drastically as more TFs are involved. To make it more computationally feasible, we converted the full model into a pairwise-logic model. The pairwise-logic model only identifies the dominant two-TF logics and thus can greatly reduce the model's complexity.

To test the performance of LogicTRN, we conducted a simulation study. The simulation platform provides us a fully controlled environment to assess the accuracy of network reconstruction. In this simulation, we repeated the experiments with random settings in each run to reconstruct the TRN. The overall results with 1000 runs suggest that our method is very accurate and reliable in identifying the regulatory logics. We next compared LogicTRN with four existing algorithms of TF target identification[4-6, 8, 13] using one real-world dataset. LogicTRN performs the best in detection of known TF target genes. We applied LogicTRN to reconstruct the TRNs during E2-induced breast cancer cell development and the hiPSC-derived CM differentiation. In breast cancer, certain bioinformatics methods have tried to define interactions among ESR1, co-TFs, and genes. However, they did not elucidate logic relationships between TFs for the regulated genes. In fact, these methods are also based on the integration of gene expression data and ChIP-seq-binding peaks or binding sites of TFs for qualitative identification of target gene. For example, Expectation Maximization of Binding and Expression pRofiles employed an unsupervised machine learning algorithm to infer the gene targets of sets of ESR1 and co-TFs[48]. Using a Bayesian multivariate modeling approach, one can reveal the dynamic properties of the ESR1-centered regulatory network and associated distinct biological functions[30]. Similarly, a logistic regression model coupling with dynamic Bayesian network was used to construct the dynamic networks at four developmental stages from mouse ESC to CM through combining RNA-seq and ChIP-seq data of cardiac TFs[49]. LogicTRN establishes a strong link with regulatory logics of ESR1 and GATA3 in E2-induced breast cancer cell development or MESP1 in the hiPSC-derived CM differentiation, as well as their related pathways or functional processes, in agreement with viewpoints previously drawn from biological experiments. The result from both computational modeling and experimental consistency suggests that LogicTRN is suitable for decoding co-regulatory features of TFs and their target genes in other biological systems. To our knowledge, this is the first use of regulatory logics for uncovering dynamic transcriptional gene regulation in a high-throughput manner.

In summary, LogicTRN is a systematic approach developed based on the well-established theories of *cis*-regulatory logics and transcriptional kinetics, which provides an efficient way to directly infer TF–TF interactions in regulating target genes. Through simulation and comparison studies, our model show its reliability and robustness in capturing TF regulatory logics and identifying targets of TFs, and in constructing TRNs. Applications in human breast cancer and the hiPSC-derived CM development demonstrate consistency between the identified TF gene regulation and the previous experiments. With increasing number of dynamic high-throughput data of gene expression and multiple regulator binding signals, our method would provide a powerful tool in interpreting the dynamics and complexity of transcriptional gene regulation. In the future, it should be possible to extend our method to cover gene regulation by other

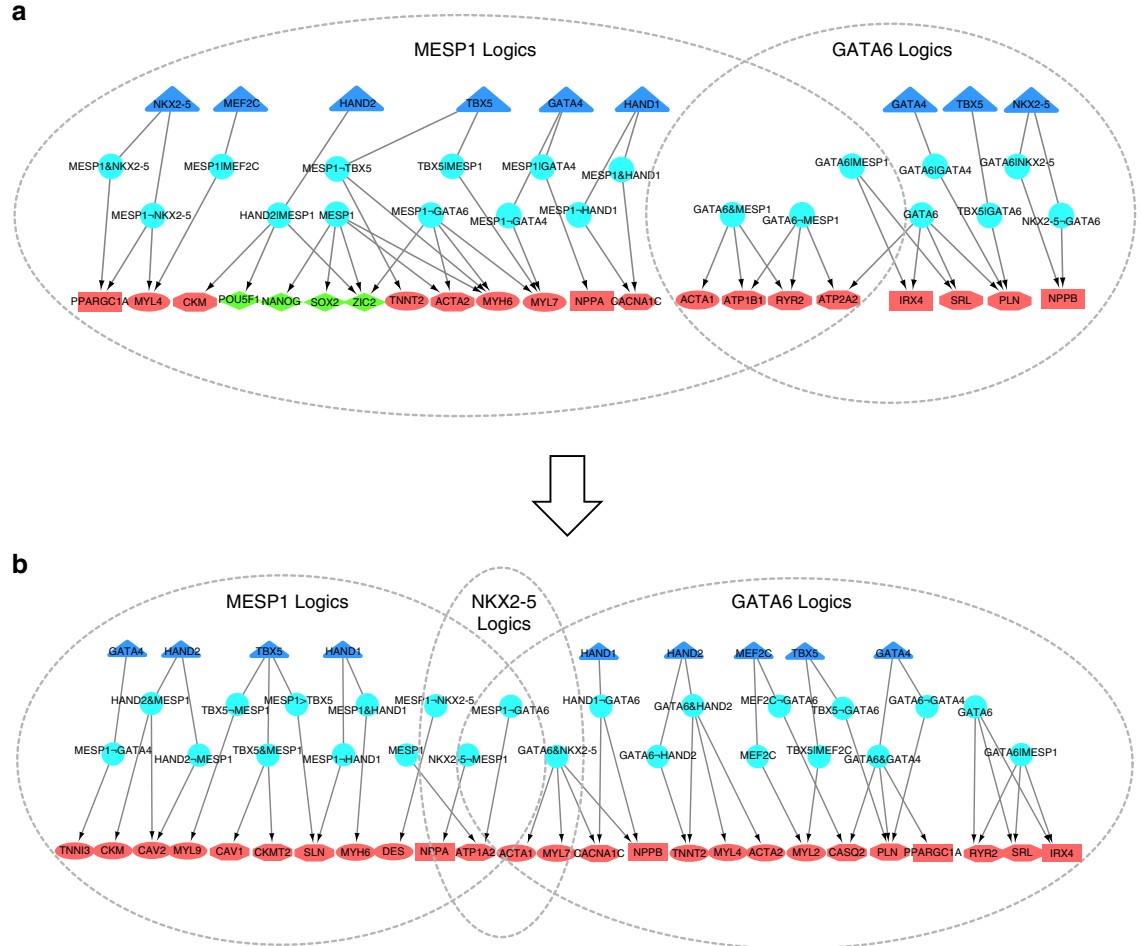

**Fig. 6** The TRNs of hiPSC-derived CM differentiation. Time-course gene expression microarray data was employed. The data represents 0, 3, 7, 10, 14, 20, 28, 35, 45, 60, 90, and 120 days following the hiPSC-derived CM differentiation, which was divided into four stages, T1 (0–3 days), T2 (7–20 days), T3 (28–45 days), and T4 (60-120 days). For **a** (at T1-T2 stage) and **b** (at T2-T3 stage), triangle nodes in blue refer to TFs, and triangle nodes in light blue are TF logics. The target genes in different pathways of logics are represented in oval nodes (contraction), diamond nodes (stem cell pluripotency), hexagon nodes (Ca handling), and rectangle nodes (heart development). Nodes in red and green represent upregulated and downregulated genes, respectively. Logic relationships include AND (&), NOT (¬), and OR (I)

regulators, such as epigenetic regulators, and other DNA-binding proteins. Such an integrated systems biology approach should help researchers develop accurate model transcriptional relationships among molecules in a biological system as a whole, thus permitting greater understanding of complex biological processes associated with development, aging, and disease.

## Methods

**Transcriptional kinetics of regulatory logics.** In study of transcriptional kinetics, the concentration of mRNA is often considered as the result of two opposite processes: gene synthesis and degradation[50]. When RNA polymerase binds to the promoter of the target gene, transcripts are initiated. If all the newly synthesized pre-RNAs can become mature messengers, then the rate of mRNA synthesis should be equal to the rate of transcript initiation. Therefore, the change of mRNA expression during a time interval equals to the rate of transcript initiation, minus the rate of mRNA degradation. Let $y_m(t)$ represent the gene expression at time $t$, the transcriptional kinetics can be expressed as an ordinary differential equation (ODE)[51]:

$$\frac{dy_m(t)}{dt} = I_s(t) - k_{dm}y_m(t) \tag{1}$$

where $I_s(t)$ is transcript initiation rate, and $k_{dm}$ denotes mRNA degradation rate which is normally treated as a constant during a process. The transcript initiation rate, however, is controlled by TF-DNA binding occupancy, which is defined as the probability that the gene promoter sites are occupied by TFs. Let $Y(t)$ denote the binding occupancy of a TF at time $t$ to its target gene, then $I_s(t)$ can be represented

as the regulatory function below[51]:

$$I_s(t) = I_{max}\left(1 - \exp\left(-\frac{k_b Y(t - T_m)}{I_{max}}\right)\right) \tag{2}$$

where $I_{max}$ is the physical limit of transcript initiation rate determined by the RNA elongation speed and the size of the polymerase, $k_b$ is the TF activation strength, and $T_m$ the transcriptional delay.

When two TFs are engaged in regulating a gene, the TFs might interact with each other in different ways. TF interactions are often represented as basic logics including AND, OR, and NOT[15, 52–54]. The AND logic describes the situation that the gene is only activated when the two TFs concurrently binding to the gene promoter, and OR logic represents that the gene can be independently activated by either of the two TFs, while NOT logic characterizes the inhibitive operation. Transcriptional initiation under different logics is expressed as different functions. The regulatory function of AND and OR logics can be written as below[51]:

**AND:** A&B     $I_s(t) = I_{max} \cdot \left(1 - \exp\left(-\frac{k_{bAB}}{I_{max}}Y_A(t - T_m) \cdot Y_B(t - T_m)\right)\right)$ (3)

**OR:** A|B

$$I_s(t) = \frac{I_{max}}{2} \cdot \left\{\left(1 - \exp\left(-\frac{k_{bA}}{I_{max}}Y_A(t - T_m)\right)\right) + \left(1 - \exp\left(-\frac{k_{bB}}{I_{max}}Y_B(t - T_m)\right)\right)\right\} \tag{4}$$

Note both AND and OR logic satisfies the commutative law, which indicates that switching of the sequence of operators does not change the meaning of the logic.

The NOT logic is used to represent the repressive regulation. As gene transcription in eukaryote cells is "off" by default, so for any expressed genes, there must be at least one activator in its regulatory logic. Repressor TFs modulate gene expression by interfering or inhibiting binding of the activator TFs. Thereby the regulatory function of NOT logic with an activator $A$ and a repressor $B$ can be written as:

$$\textbf{NOT}: \quad A\neg B \qquad I_s(t) = I_{\max} \cdot \left(1 - \exp\left(-\frac{k_{bA}}{I_{\max}} Y_A(t - T_m) \cdot (1 - Y_B(t - T_m))\right)\right) \tag{5}$$

Note the NOT logic does not satisfy the commutative law, i.e., $A\neg B$ and $B\neg A$ are different logics.

**Polynomial functions of regulatory logics.** The nonlinear regulatory functions in (2)–(5) make it difficult to computationally identify the model. To address this issue, we apply Taylor expansion on the nonlinear regulatory function and obtain:

$$I_s(t) = I_{\max} \sum_{n=1}^{\infty} \left((-1)^{n+1} \left(\frac{k_b}{I_{\max}}\right)^n \cdot Y^n(t - T_m)/n!\right) \tag{6}$$

where $n$ is the order of Taylor expansion. Substituting (6) into (3)–(5) and omitting the term of $(t - T_m)$ for simplification purpose, we obtain the polynomial regulatory functions:

$$\textbf{AND}: \quad A\&B \qquad I_s(t) = I_{\max} \sum_{n=1}^{\infty} \left((-1)^{n+1} \left(\frac{k_{bAB}}{I_{\max}}\right)^n \cdot Y_A^n \cdot Y_B^n/n!\right) \tag{7}$$

$$\textbf{OR}: \quad A|B \qquad I_s(t) = (I_{\max}/2) \cdot \sum_{n=1}^{\infty} \left((-1)^{n+1} \left(\frac{k_{bA}}{I_{\max}}\right)^n \cdot Y_A^n + (-1)^{n+1} \left(\frac{k_{bB}}{I_{\max}}\right)^n \cdot Y_B^n/n!\right) \tag{8}$$

$$\textbf{NOT}: \quad A\neg B \qquad I_s(t) = I_{\max} \sum_{n=1}^{\infty} \left((-1)^{n+1} \left(\frac{k_{bA}}{I_{\max}}\right)^n \cdot Y_A^n \cdot (1 - Y_B)^n/n!\right) \tag{9}$$

**Multi-TF regulatory function.** The binary logic between two TFs can be extended to combinatorial logic to represent interactions among multiple TFs. For example, "A&B&C" is used to describe that a gene is only activated when the TFs A, B, and C are concurrently binding to the gene's promoter. In fact, there could be unlimited number of combinatorial logics for multiple TFs, if a TF can be used repeatedly. Here, we define the concept of URL, which restrict that a TF can at most occur once in the logic. For instance, "A", "A|B" and "A&B|C", are legal URLs, whereas "(A&B)|B", "A¬A", and "A&B&C|C" are illegal because they include duplicate TFs. According to URL, one can obtain a closed set of combinatorial logics for a group of TFs. For instance, two TFs can form total six URLs (Supplementary Methods). The number of URLs for a group of TFs can be computed according to Supplementary Methods.

For a target gene, we assume that each of the URLs has a probability of being involved in its regulation. Let $R_i$ represent the $i$th URL, $\omega_i$ be the probability of $R_i$, and $I_i$ be the contribution of $R_i$ to gene initiation. Then, the overall gene initiation by all the URLs can be expressed as:

$$I_s = \sum_i^{N_R} \omega_i I_i \tag{10}$$

where $N_R$ is the number of URLs for this target gene.

Let $Y_1, Y_2, \cdots, Y_p$ represent the TF-DNA-binding occupancies of a number of $p$ TFs on the target gene. We first define a general composite variable, $Z_{i_1, \cdots i_p} = Y_1^{i_1} Y_2^{i_2} \cdots Y_p^{i_p}$, where $i_1, \cdots i_p \in [0, n]$, and $n$ is the order of Taylor expansion in the polynomial regulatory function. Then, the set of composite variables, $\mathbf{Z}$, can be expressed as $\{Z_{i_1, \cdots i_p} = Y_1^{i_1} Y_2^{i_2} \cdots Y_p^{i_p} | i_1, \cdots i_p \in [0, n]\}$. We found that the regulatory function of a URL, regardless its specific TFs or logics, can be generally expressed as:

$$I_i = \sum_{i_p=0}^{n} \cdots \sum_{i_2=0}^{n} \sum_{i_1=0}^{n} \left(a_{i_1, i_2, \cdots i_p} \cdot Z_{i_1, \cdots i_p}\right) + \varepsilon_i \tag{11}$$

where $a_{i_1, \cdots i_p}$ is the coefficient associating with the composite variable $Z_{i_1, \cdots i_p}$, and $\varepsilon_i$ is the truncation error of Taylor series. Equation (11) means that the regulatory function of a URL can be converted to a polynomial function of a set of common composite variables. For simplification purpose, we re-organize $\{Z_{i_1, \cdots i_p} = Y_1^{i_1} Y_2^{i_2} \cdots Y_p^{i_p} | i_1, \cdots i_p \in [0, n]\}$ to be $\{Z_j | j \in [1, N_Z]\}$, in which $Z_j$ represents the $j$th element in the set, and $N_Z$ is the number of elements. Using $Z_j$ to replace $Z_{i_1, \cdots i_p}$, and $a_{ij}$ to replace $a_{i_1, i_2, \cdots i_p}$, then (11) can be re-written as:

$$I_i = \sum_{j=1}^{N_Z} \left(a_{ij} \cdot Z_j\right) + \varepsilon_i \tag{12}$$

The coefficient $a_{ij}$ is associated with the $i$th URL and the $j$th composite variable, which in fact is a function of the kinetic parameters (including the TF regulatory

strength and maximal initiation rate). A further elaboration using two-TF regulation as an example can be found in Supplementary Methods.

**The model equation of transcriptional gene regulation.** Substitute (12) into (10), let $\beta_j = \sum_i^{N_R} \omega_i a_{ij}$ and $\varepsilon = \sum_{i=1}^{N_Z} \omega_i \varepsilon_i$, then we have

$$I_s = \sum_{j=1}^{N_Z} \beta_j \cdot Z_j + \varepsilon \tag{13}$$

Here $\beta_j$ is a function of the kinetic parameters and probability $\omega_i$, while $\beta_j$ can only be specified when $p$ (the number of regulators) and $n$ (the degree of Taylor expansion) are giving.

Substitute (13) into (1), and convert the differential equation to a difference equation, we obtain the model equation of transcriptional gene regulation:

$$\hat{y}_m(t_l) = \sum_{j=1}^{N_Z} \left(\beta_j \cdot \Delta t\right) \cdot Z_j + (1 - k_{dm} \cdot \Delta t) \cdot y_m(t_{l-1}) + \varepsilon \cdot \Delta t \tag{14}$$

where $l = 2, \cdots, L$ are the indices of time points, $b_j = \beta_j \cdot \Delta t$ and $b_0 = 1 - k_{dm} \cdot \Delta t$ are the regression coefficients, $\varepsilon' = \varepsilon \cdot \Delta t$ is the truncation error, and $\Delta t = t_l - t_{l-1}$ is the sampling time interval. In case that the samples are acquired with equal time interval, then $\Delta t$ can be treated as a unit value and replaced by 1. Therefore, the model equation in (14) can be simplified as

$$\hat{y}_m(t_l) = \sum_{j=1}^{N_Z} \beta_j \cdot Z_j + (1 - k_{dm}) \cdot y_m(t_{l-1}) + \varepsilon \tag{15}$$

The model equation describes the relationships between dynamic gene expression and TF interactions. Theoretically, once the coefficients ($\beta$'s) are solved out from the model equation, the regulatory logics and kinetic parameters can thus be determined accordingly.

**The pairwise-logic model equation.** Since the number of URLs increases drastically with the number of TFs (Supplementary Methods), it is not feasible to computationally solve the model when there are multiple TFs. Here propose to convert the full model to be a pairwise-logic model, such that to greatly reduce the number of URLs and the model's complexity. The TF pairs of a group of $p$ TFs $\{A_u, u = 1, p\}$ can be expressed as $\{(A_u, A_v); u \in [1, p]$ and $v \in [u + 1, p]\}$. For each TF pair, the URLs and the composite variables are countable according to Supplementary Methods. The contribution of a TF pair to gene initiation can be expressed as:

$$I^{(u,v)} = \sum_{j=1}^{N_{Z^{(u,v)}}} \beta_j^{(u,v)} \cdot Z_j^{(u,v)} \tag{16}$$

Let $q_{uv}$ denote the probability involving the TF pair $(A_u, A_v)$ in gene regulation, the gene regulatory function can be represented as:

$$I_s = \sum_{u=1}^{p} \sum_{v=u+1}^{p} q_{uv} \cdot I^{(u,v)} \tag{17}$$

Substitute (16) in (17), we thus obtain:

$$I_s = \sum_{u=1}^{p} \sum_{v=u+1}^{p} \left(q_{uv} \cdot \sum_{j=1}^{N_{Z^{(u,v)}}} \beta_j^{(u,v)} \cdot Z_j^{(u,v)}\right) \tag{18}$$

Let $\mathbf{Z}^{(\text{all})} = \{Z_j^{(u,v)} | u \in [1, p], v \in [u + 1, p], j \in [1, N_{Z^{(u,v)}}]\}$ represent the set of composite variables of all TF pairs, where $N_Z^{(\text{all})}$ is the number of elements in $\mathbf{Z}^{(\text{all})}$. Then $\mathbf{Z}^{(\text{all})}$ can be simplified to $\{Z_j^{(\text{all})} | j \in [1, N_Z^{(\text{all})}]\}$, and the coefficient set $\boldsymbol{\beta}^{(\text{all})} = \{\beta_j^{(u,v)} | u \in [1, p], v \in [u + 1, p], j \in [1, N_{Z^{(u,v)}}]\}$ can be simplified to $\{\beta_j^{(\text{all})} | j \in [1, N_Z^{(\text{all})}]\}$. Therefore, (18) can be re-written as:

$$I_s = \sum_{j=1}^{N_Z^{(\text{all})}} \beta_j^{(\text{all})} \cdot Z_j^{(\text{all})} \tag{19}$$

Substituting (19) into (1), we can obtain the pairwise-logic model equation:

$$\hat{y}_m(t_l) = \sum_{j=1}^{N_Z^{(\text{all})}} \beta_j^{(\text{all})} \cdot Z_j^{(\text{all})} + (1 - k_{dm}) \cdot y_m(t_{l-1}) + \varepsilon \tag{20}$$

The pairwise-logic model uses much less URLs. For instance, in case of five TFs, there are 351,816 URLs in the full model, while only 66 URLs in the pairwise-logic model. Now the problem is how to determine the dominant regulatory logic and kinetic parameters from experimental data. Here, we introduced a probabilistic approach to solve the model equation, details are presented in Supplementary Methods. The algorithm of LogicTRN is presented in the Supplementary Methods.

**Comparison among different methods.** To compare the performance of different methods in identifying the target genes of TFs, we conducted widely-used ROC and PR analyses, respectively. The ROC analysis plots a two-dimensional curve between TPR and FPR of the predictions, whereas PR curves are to plot precision against recall. Computational algorithms or methods can be evaluated by analyzing the AUC of ROC, and PR curves. A high AUC means the better performance of an algorithm or method in identifying TF target genes. LogicTRN and four methods COGRIM[8], APG[55], NCA[4], and PTHGRN[13], were separately applied to analyze the input data (Supplementary Note 1). First, we have generated a benchmark set consisting of a "positive" and a "negative" set. To define the positive set, we derived the known target genes of the four mouse ESC TFs (Oct4/Pou5fl, Sox2, Nanog, and Suz12) from our web server Chip-Array2[56] that is based on ChIP-seq-binding peaks and experimental testing by TF knockdown or overexpression

(Supplementary Data 3). A set of genes in Supplementary Data 3 were used as positive signals if they are present in Affymetrix Mouse Expression 430A Array and also contain ChIP-seq-binding loci of the four TFs based on binding peak analysis (Supplementary Data 2). The negative set includes those genes that are not positives but present in both Supplementary Data 2 and the 430A Array (see Supplementary Data 3). They represent genome-wide unknown-binding target genes of the four TFs. The predicted target genes using the five methods were listed in Supplementary Data 3. To conduct an unbiased evaluation for ROC and PR analyses, we utilized full size of positives and negatives. Through comparison of identification of known TF target genes, we are able to assess the performance of the five methods.

**Validation of predicted TF logics**. To validate the logics predicted by LogicTRN, we analyzed the influence of logics to their target genes obtained after knockdown or overexpression of the engaged TFs. In breast cancer, we analyzed the data set (GEO accession number GSE31912) that accounts for gene expression profile in MCF7 breast cancer cells after knockdown of ESR1, GATA3, and CEBPB (Results see Supplementary Fig. 3). We used the violin plot to show the density distribution of the expression changes of the target genes regulated by a TF logic. The differences of the distribution of expression change between each group of target genes were determined by one-way analysis of variance (ANOVA) using IBM SPSS Statistics 23. A two-tailed $p$-value was calculated by ANOVA and corrected by false discovery rate. A $p$-value of less than 0.05 was considered statistically significant, and indicates that the distribution of target genes of a logic formed by two TFs are influenced significantly by TF knockdown comparing to other logics of the two TFs. Thus, the TF logic is likely to be true. Similarly, to validate the logics of hiPSC-CM differentiation, we used the Mesp1 overexpression-treated gene expression data (GEO database GSE5976). Same analytical approaches were adopted. The results were shown in Supplementary Fig. 6.

**Code availability**. The LogicTRN codes used in this study are available at http://staffweb.hkbu.edu.hk/hlzhu/2017LogicTRN_codes.html or upon request.

**Data availability**. Microarray and ChIP-seq data were published previously and are available on GEO (Comparison study: GSE3231 and GSE11724. Application Study for breast cancer cells: GSE26831, GSE29073, GSE32692, GSE32465, GSE14664, GSE19013, GSE54855, GSE40129, as well as E-MTAB-223 from ArrayExpress. Application Study for hiPSC-derived CM: GSE35671. Validation of predicted TF logics using knockdown or overexpression of TFs: GSE31912 and GSE5976. All other remaining data are available in the Article and Supplementary Files, or available from the authors upon request.

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

## Acknowledgements

This work was supported by the Research Grants Council of Hong Kong: GRF (grant No. 212111, 212613, and 17100214) and Theme-based Research Scheme T13-706/11; Interdisciplinary Research Matching Scheme of Hong Kong Baptist University RC-IRMS/12-13/02RC-IRMS/13-14/03 RC-IRMS/14-15/02; Strategic Development Fund of Hong Kong Baptist University SDF13-1209-P01; Faculty Research Grants of Hong Kong Baptist University FRG2/16-17/052; Seed Funding for Basic Research of the University of Hong Kong, RCGAS0768448734.

## Author contributions

H.Z. developed the modeling theory and the algorithms. B.Y., D.G., B.H., C.W. and J.Q. performed data processing and analysis. H.Z., B.Y., D.G., J.W., K.R.B., G.Z. and A.L. participated in drafting the manuscript. All authors read and approved the final manuscript.

## Additional information

**Competing interests:** The authors declare no competing financial interests.

