## [Peer Review file · Nature Communications]

Reviewers' comments:

Reviewer #1 (Remarks to the Author):

A. summary of key results

The article "Identifying Logics for Inferring Transcriptional Regulatory Networks" by Yan et al. proposes a novel approach to determine gene regulatory networks from expression and TF binding data. In addition to previously proposed methods, the logicTRN approach also determines the regulatory logic and the combinatorial aspect of co-regulation by several TFs. The method is validated on simulated datasets, and the authors also apply their method to several real datasets and show that the regulatory logic (i.e. the combinations of TFs) correspond to literature validated combinations.

B. Originality and interest

In principle, this method represents an attractive advance compared to the large number of already available methods published in the past. Being able to distinguish the role of some TFs in the co-regulation of target genes is interesting in particular to predict downstream effects of interventions, for example therapeutic interventions.

C. Quality of presentation

Overall, there is a lack of clarity in the presentation of the details. I acknowledge that the authors have tried to present details of the methods in the supplementary documents, but there are a number of shortcuts that make the understanding of the method challenging:

- no details are given on the way the kinetic parameters are estimated, apart from a vague reference to usage of a genetic algorithm (end of supp. method 3 and 4)
- notations are not consistent: gene expression is referred to as x in supp. method 5, whereas it is y_m in all other parts of the manuscript
- the estimation of the a parameter in supp method 5 is very obscure, and should be detailed as this is a key aspect of the whole method.

The writing should be improved, and there are numerous typos all over the manuscript which should be corrected.

D. Appropriate use of statistics and treatment of uncertainties

not relevant in this case

E. Conclusions: robustness, validity, reliability

See suggestions below

F. Suggested improvements: experiments, data for possible revision

However, there are a number of points which should be addressed by the authors before acceptance of the manuscript could be considered.

1. Since the methods is based per construction on binding data (either motifs of ChIP-seq data), I wonder to what extend the regulatory logic is already conditioned by this binding

information; for example, if many genes show binding for TF1 and TF2 in their promoters as determined by ChIP-seq data, then it is expected that the logic TF1 & TF2 will appear as the most likely logic. On the other hand, if we have in most of the promoters either TF1 or TF2, then the logic TF1 | TF2 should be the most likely. So it is not completely clear to me what the methods adds to this "naive" approach. I see that the relative contributions of the URL can be quantitatively assessed, but could the overall logic not already be obtained from naive considerations ?

2. Related to the previous comment : most of the methods that logicTRN is compared to do not rely on binding data to infer the regulatory network, but only use expression data. Hence, it is somewhat unfair to compare the methods by counting the number of recovered target genes for the transcription factors, since apparently logicTRN has this information built-in by requiring the binding data. Either this should be made more clear in the description of the comparison design, or the comparison should be modified and another criteria be used for the comparison.

3. The authors determine the enrichment of the genes targeted by the high ranking regulatory logic among the differentially expressed genes. I would strongly recommend a negative control, for example by focusing on regulatory logic involving the same transcription factors, but which are NOT among the enriched URLs, and check the level of enrichment of the target genes among the differentially expressed genes. I would be curious to see a difference in the enrichment between high ranking and low ranking URLs.

4. A more convincing validation would be to use a datasets obtained after inhibition or over-expression of one of the TFs involved in combinatorial regulation, and checking whether the target genes predicted to be affected by URLs involving this TF react as predicted by the model; for example, in the breast cancer dataset, ESR1 is sometimes predicted to be positively associated with some other TFs (ESR1 & GATA3), and sometimes associated with the OR logic (ESR1 OR CEBPB). Predicted target genes in the first case should be more affected in case of ESR1 knock-down than target gene in the second case. Since many such datasets are available (Theodorou et al. have such datasets available for example), it should be possible to make such a validation.

G. References: appropriate credit to previous work?

Some additional references could be added in the introduction with respect to methods for TRN inference; for example, the DREAM challenges (DREAM4 and DREAM5) which have compared and benchmarked such methods should be referred to.

H. Clarity and context: lucidity of abstract/summary, appropriateness of abstract, introduction and conclusions

See previous comments; some of the technical aspects should be detailed

Reviewer #2 (Remarks to the Author):

The authors describe an approach that takes into account time series gene expression data and ChIP seq experiments to infer gene regulatory networks. The special feature of the presented approach is the simultaneous inference of the boolean regulatory logics by which sets of TFs regulate their respective target genes. This is an important methodological asset as its not adequate to model eucaryote gene expression via simple binary TF-target interactions to explain gene expression. For runtime reasons, boolean combinatorial logics is restricted to pairs of TFs. In general the paper is well written, although some information is spread over several sections and not well enough specified in a single place, see below for details. The methods to the extend they

are explained in the main paper are lucid. But some important methods are omitted from the main paper, making it difficult to assess the merits of logicTRN. References seem appropriate, the abstract is well written.

Major compulsory comments

The simulation model is only explained in the supplement. As assessment of logicTRNs ability to re-infer simulated networks, this is one of the two major indications of the approaches power. Importantly, the simulation approach must be clarified in the main manuscript to the extent to demonstrate that it does not give any unfair advantage to logicTRN

Similarly, in the section "Comparison with other methods" the datasets, the gold standards and the exact mode of assessment must be clarified (not explained anywhere in the main manuscript, again only supplement). Without specifying the assessment strategy the obtained results cannot be interpreted.

Minor comments

p3 may not necessarily have physical meanings
What does it mean, please clarify

p3 DNA binding TFs are not always involved in the regulation of gene transcription
This is too general, please specify

p4 to quantitatively identify the TF regulatory logics and their target genes so as to reconstruct TRNs
Its not clear what it means

p5 Successful application of LogicTRN shows its ability to accurately reconstruct TRNs, and to describe the dynamic transcriptional profiles.
Also not clear, where do you evaluate the dynamic transcriptional profiles in the paper?

p6 Based on the two types of input data, we can calculate the TF-DNA binding occupancy (C), the details can be found in Supplemental Method Five (Supplementary Information).
This should also be clarified in the main manuscripts methods section

p6 a series of model equations can be formed at each time point
Please clarify here the significance of the model equations

p6 Thus, we are able to put together a model equation to connect the gene expression and TF-DNA binding occupancy.
In combination with the previous comment, please be more specific on how model equations connect the gene expression and TF-DNA binding occupancy

p7 To evaluate the performance of LogicTRN, we constructed a simulation platform including six nodes, where each node represents a TF gene. Detailed configuration of this platform can be found in Supplemental Method Six (Supplementary Information).
This explanation is not comprehensive enough, see also first major comment.

p9
Why is AUC used for the simulation study, but not here? What is the intuitive meaning of a certain SNR here? Please assess performance systematically, and also use the area under the precision recall curve

p9 TFs that can physically bind and access target genes while we are constructing the model equations

Do you mean you restrict the logic to ChIP interactions?

p15 Unlike existing integrative analyses that treat gene expression and TF-DNA binding as two separate processes, our approach puts these two types of data into one single model, which is more biologically meaningful.

This statement does not explain well enough how the integration takes place. To me it was necessary to read this, the first result section and the methods to get enough understanding

p16 our method established a biological model which treats gene expression and TF-DNA binding as indivisible aspects of the gene transcription.

Repetition

p17 in agreement with many conclusions previously drawn from the biologic experiments. The information from both computational and literature search

"many conclusions previously drawn" and "both computational and literature search" are not clear

p17 Our method can be extended to the examination of gene expression regulated not only by TFs, but also miRNA, epigenetic regulators, or other factors. It can also be applied to dissect the roles of DNA or RNA binding proteins whose functions are not determined yet, or unknown cis-regulatory elements.

What of this is special to your methods?

p17 Comparison with existing methods, our model not only exhibits detailed TF co-regulation patterns,

What "TF co-regulation patterns" are you referring to?

Typos etc

p4 to develop a transcriptional regulatory model, LogicTRN, based which one is able to

p7 the same approaching on each gene

p15 which makes it difficult for developing

Responses to reviewers' comments or suggestions

Reviewer #1

A. summary of key results

The article "Identifying Logics for Inferring Transcriptional Regulatory Networks" by Yan et al. proposes a novel approach to determine gene regulatory networks from expression and TF binding data. In addition to previously proposed methods, the logicTRN approach also determines the regulatory logic and the combinatorial aspect of co-regulation by several TFs. The method is validated on simulated datasets, and the authors also apply their method to several real datasets and show that the regulatory logic (i.e. the combinations of TFs) correspond to literature validated combinations.

B. Originality and interest

In principle, this method represents an attractive advance compared to the large number of already available methods published in the past. Being able to distinguish the role of some TFs in the co-regulation of target genes is interesting in particular to predict downstream effects of interventions, for example therapeutic interventions.

C. Quality of presentation

Overall, there is a lack of clarity in the presentation of the details. I acknowledge that the authors have tried to present details of the methods in the supplementary documents, but there are a number of shortcuts that make the understanding of the method challenging:

- no details are given on the way the kinetic parameters are estimated, apart from a vague reference to usage of a genetic algorithm (end of supp. method 3 and 4)

#Response: Thank you for this suggestion. In the revision, we removed the part of kinetic parameter estimation, such that to make this manuscript focus more on logic identification.

- notations are not consistent: gene expression is referred to as x in supp. method 5, whereas it is y_m in all other parts of the manuscript

#Response: The reviewer is absolutely correct, and the inconsistency of notations was indeed a point of confusion. In this revision, we have replaced x using y_{TF} to be the variable representing the TF gene expression (see Page 4 in Supplementary Information). We still use y_m to represent the gene expression of the target gene in the main manuscript (see page 6).

- the estimation of the a parameter in supp method 5 is very obscure, and should be detailed as this is a key aspect of the whole method.

#Response: We thank reviewer for this valuable comment. In the revision, we provide a detailed description of estimating the parameter a in the Supplemental Method Four. Details can be found in the Supplementary Information page 5.

F. Suggested improvements: experiments, data for possible revision

However, there are a number of points which should be addressed by the authors before acceptance of the manuscript could be considered.

1. Since the methods is based per construction on binding data (either motifs of ChIP-seq data), I wonder to what extend the regulatory logic is already conditioned by this binding information; for example, if many genes show binding for TF1 and TF2 in their promoters as determined by ChIP-seq data, then it is expected that the logic TF1 & TF2 will appear as the most likely logic. On the other hand, if we have in most of the promoters either TF1 or TF2, then the logic TF1 | TF2 should be the most likely. So it is not completely clear to me what the methods adds to this "naive" approach. I see that the relative contributions of the URL can be quantitatively assessed, but could the overall logic not already be obtained from naive considerations ?

#Response: We thank the reviewer for excellent comment, and we have tried to address this point in the revision. In response, it is clear that TF-DNA binding information accommodate partial information of TF regulatory logics, which can be utilized by a "naive" approach to infer regulatory logics as mentioned by the reviewer. The "naive" approach, however, has limitations and cannot be accurate because the binding information only represents potential binding sites or strength of TF binding, which may not be functional in regulating gene transcription in a particular biological process. On the other hand, gene expression data reflects actual result of gene transcription. Therefore, by appropriately combining both data types, this algorithm is better able to identify authentic TF target genes.

Altogether, this shows that the integration of gene expression and TF-DNA binding is more powerful as comparing to the "naive" approach. Two additional points validating this conclusion are made in the following paragraphs.

1. Integrative analyses are more accurate in identifying the TF target genes comparing to the "naive" approach. To address this point, we applied all five integrative analyses and the "naive" approach on the two examples of lung cancer and the mouse embryonic stem cell (Dataset One, Supplemental Information). In Figure 1 of this letter, we show the ROC of all six methods using the two datasets. It can be clearly seen that the "naive" approach does not provide sufficient information for identifying the TF target genes, and that LogicTRN is superior than the other algorithms.

Figure 1. Comparison of five integrative analysis methods and the “naive” approach. A, ROC and AUC of the six methods on lung cancer data. B, ROC and AUC of the six methods on mouse embryonic stem cell data.

- LogicTRN can significantly reduce the predicted logics comparing to those detected by “naive” approach. To clearly illustrate this point of view, we conducted example test for TFs GATA3 and FOXM1 using breast cancer data. First, according to ChIP-seq binding data of the two TFs, we detected 6342 genes containing both binding loci using TF binding data. Thus, if we use the “naive” approach, one might find all the 6342 genes are regulated by GATA3 and FOXM1. However, using LogicTRN, we found that only 73 genes are regulated the two TFs. the resultant logics include GATA3&FOXM1 (10 genes), FOXM1–GATA3 (3 genes), GATA3–FOXM1 (29 genes), and GATA3|FOXM1 (31 genes). Therefore, LogicTRN greatly reduced the number of predicted logics, making it a useful tool in generating hypothesis for further experimental validation.

Figure 2. Total 6342 genes were detected with both GATA3 and FOXM1 binding loci by ChIP-seq. Of 73 genes were identified as TF target genes by LogicTRN, where 10 genes are regulated by GATA3&FOXM1, 3 genes by FOXM1–GATA3, 29 genes by GATA3–FOXM1, and 31 genes by GATA3|FOXM1.

2. Related to the previous comment: most of the methods that logicTRN is compared to do not rely on binding data to infer the regulatory network, but only use expression data. Hence, it is somewhat unfair to compare the methods by counting the number of recovered target genes for the transcription factors, since apparently logicTRN has this information built-in by requiring the binding data. Either this should be made more clear in the description of the comparison design, or the comparison should be modified and another criteria be used for the comparison.

#Response: We thank the reviewer for pointing out this issue. In response, we have re-illustrated in detail that all the other four methods are using both TF binding information and gene expression data (in the main manuscript see page 3). With these modifications, we believe that we have been more clear in the description of the comparison design, which will be of benefit to the readers.

3. The authors determine the enrichment of the genes targeted by the high ranking regulatory logic among the differentially expressed genes. I would strongly recommend a negative control, for example by focusing on regulatory logic involving the same transcription factors, but which are NOT among the enriched URLs, and check the level of enrichment of the target genes among the differentially expressed genes. I would be curious to see a difference in the enrichment between high ranking and low ranking URLs.

#Response: We thank the reviewer for this excellent comment. In Supplementary Information, we added Supplemental Method Seven for such the negative controls. As suggested, we constructed the negative control by collecting the logics not among the enriched URLs but have the same TFs involved. We evaluated the enrichment of the target genes among the differentially expressed genes, and then transformed the p value to negative 10-based logarithm (NBL). As shown at Figure 3, most of the negative control (blue bar) has lower NBL than the enriched URLs (orange bar). For T1-T2 and T2-T3 stages in human breast cancer and hiPSC-CM, the p values between enriched URLs and negative control was calculated, respectively by the two-tail students test. As shown in Figure 3, 3 of the 4 stages show $p < 0.05$. The testing result indicates that high ranked logics determined by our method can recruit more differentially expressed target genes.

A

p value = 0.032

B

p value=0.009

C

p value=0.202

D

p value=0.012

Figure 3. Comparison of TF logic target genes between the enriched URLs and the “Negative control” in human. A (T1-T2 stage) and B (T2-T3 stage) are human breast cancer data, C (T1-T2 stage) and D (T2-T3 stage) are hiPSC-CM data. Orange and blue bars represent enriched URLs and “Negative control”, respectively.

4. A more convincing validation would be to use a datasets obtained after inhibition or over-expression of one of the TFs involved in combinatorial regulation, and checking whether the target genes predicted to be affected by URLs involving this TF react as predicted by the model; for example, in the breast cancer dataset, ESR1 is sometimes predicted to be positively associated with some other TFs (ESR1 & GATA3), and sometimes associated with the OR logic (ESR1 OR CEBPB). Predicted target genes in the first case should be more affected in case of ESR1 knock-down than target gene in the second case. Since many such datasets are available (Theodorou et al. have such datasets available for example), it should be possible to make such a validation.

#Response: We thank the reviewer’s excellent suggestion. We have now added two data showing the validation testing. The results are consistent with our model findings and strongly support the value of logicTRN for these analyses. Results page 11, 13 and Methods page 23 described the details.

It includes: 1) After knockdown of three TFs ESR1, GATA3, and CEBPB, we analyzed the influence of various logics on their target genes in breast cancer. The result was displayed in Fig.S3 of the revised manuscript. 2) By examining the influence of TF (MESP1) overexpression, we validated the target genes of MESP1–GATA6, and MESP1&HAND1 are more affected by the overexpression. The result was displayed in Fig. S6 of the revised manuscript.

G. References: appropriate credit to previous work?

Some additional references could be added in the introduction with respect to methods for TRN inference; for example, the DREAM challenges (DREAM4 and DREAM5) which have compared and benchmarked such methods should be referred to.

#Response: We thank the reviewer for pointing out this issue. We have added references about methods in the DREAM challenges in the Introduction (see page 3).

Reviewer #2

The authors describe an approach that takes into account time series gene expression data and ChIP seq experiments to infer gene regulatory networks. The special feature of the presented approach is the simultaneous inference of the boolean regulatory logics by which sets of TFs regulate their respective target genes. This is an important methodological asset as it is not adequate to model eukaryote gene expression via simple binary TF-target interactions to explain gene expression. For runtime reasons, boolean combinatorial logics is restricted to pairs of TFs. In general the paper is well written, although some information is spread over several sections and not well enough specified in a single place, see below for details. The methods to the extend they are explained in the main paper are lucid. But some important methods are omitted from the main paper, making it difficult to assess the merits of logicTRN. References seem appropriate,

the abstract is well written.

Major compulsory comments

The simulation model is only explained in the supplement. As assessment of logicTRN's ability to re-infer simulated networks, this is one of the two major indications of the approaches power. Importantly, the simulation approach must be clarified in the main manuscript to the extent to demonstrate that it does not give any unfair advantage to logicTRN.

#Response: We thank reviewers for this good suggestion. We have re-illustrated the details of simulation study in the Results of the main manuscript in attempt to better explain the approach to a wide reading audience (see page 6). In the simulation study, we employed three approaches to make sure the assessment has been fairly conducted:

1. Use canonical ODE functions and kinetic parameters to describe the dynamics of gene transcription and protein translation, instead of choose the kinetic functions that are more favorable to our methods.
2. Randomly choose the regulatory logics from all the logic sets for each gene in each run of experiment, to avoid unfair favor to LogicTRN on some particular logics.
3. Randomly set the kinetic parameters in the kinetic functions to generate TF-DNA binding occupancy and gene expression data. The random setting of parameters can avoid unfair favor to LogicTRN on some particular values.

Based on these approaches, we neither believe nor can we demonstrate that the simulation gives any unfair advantage to LogicTRN.

Similarly, in the section "Comparison with other methods" the datasets, the gold standards and the exact mode of assessment must be clarified (not explained anywhere in the main manuscript, again only supplement). Without specifying the assessment strategy the obtained results cannot be interpreted.

#Response: We thank the reviewer again for the good suggestion. We have re-evaluated the performance of all the five methods using the gold standard of ROC and AUC in the comparison study (see page 8).

Minor comments

p3 may not necessarily have physical meanings What does it mean, please clarify

#Response: In the introduction, we have clarified the section TF gene regulation. So we removed this expression.

p3 DNA binding TFs are not always involved in the regulation of gene transcription This is too general, please specify

#Response: Agreed, and we have removed this statement.

p4 to quantitatively identify the TF regulatory logics and their target genes so as to reconstruct TRNs. Its not clear what it means

#Response: We have revised the texts in Introduction.

p5 Successful application of LogicTRN shows its ability to accurately reconstruct TRNs, and to describe the dynamic transcriptional profiles.

Also not clear, where do you evaluate the dynamic transcriptional profiles in the paper?

#Response: We have removed this unclear expression and improved our explanations.

p6 Based on the two types of input data, we can calculate the TF-DNA binding occupancy (C), the details can be found in Supplemental Method Five (Supplementary Information page 5).

This should also be clarified in the main manuscripts methods section

#Response: We thank the reviewer for pointing this out. We have re-organized this part in Supplemental Method Four (Page 6 in Supplementary Information). The reason that we still leave it in Supplementary Information is that the calculation of TF-DNA binding occupancy is a peripheral procedure as can be seen in Fig. 1 in the main manuscript. In fact, this method of TF-DNA binding occupancy presented here can be replaced by other relevant approaches that won't affect the use of LogicTRN. We have, therefore, left the explanations/method of estimating TF-DNA binding occupancy in the supplemental methods, as addition to the main text made it too long for submission.

p6 a series of model equations can be formed at each time point

Please clarify here the significance of the model equations

#Response: We thank the reviewer for pointing it out. In response, we have re-illustrated that the model equations represent the transcriptional regulatory model of LogicTRN. The revision of this part now is as follows: "Third, the transcriptional regulatory model in LogicTRN is represented with a group of model equations with the inputs of gene expression data and TF-DNA binding occupancies. Solving the model equations can lead to identification of the regulatory logics of the target gene TG" (see page 6 in the main manuscript).

p6 Thus, we are able to put together a model equation to connect the gene expression and TF-DNA binding occupancy.

In combination with the previous comment, please be more specific on how model equations connect the gene expression and TF-DNA binding occupancy

#Response: We have removed this unclear expression.

p7 To evaluate the performance of LogicTRN, we constructed a simulation platform including six nodes, where each node represents a TF gene. Detailed configuration of this platform can be found in Supplemental Method Six (Supplementary Information).

This explanation is not comprehensive enough, see also first major comment.

#Response: We thank the reviewer again for the good comment. As this was central to the points of the paper, we have incorporated the simulation study and provided necessary details in the main manuscript (see page 6 in the main manuscript).

p9

Why is AUC used for the simulation study, but not here? What is the intuitive meaning of a certain SNR here? Please assess performance systematically, and also use the area under the precision recall curve

#Response: Thanks once again for the good suggestion. In response, we have re-evaluated the performance of the methods using ROC and AUC both in simulation and comparison studies (see Results section).

p9 TFs that can physically bind and access target genes while we are constructing the model equations Do you mean you restrict the logic to ChIP interactions?

#Response: We have removed this unclear expression. We prefer ChIP TF-DNA interactions (as demonstrated with the application of E2-induced breast cancer development) as TF binding signals because ChIP-seq experiments are based on cell specific and is naturally more reliable in describing the TF-DNA binding comparing to binding motifs. However, there is no problem of using LogicTRN on binding motifs data (as demonstrated in the application of hiPSC-derived CM differentiation).

p15 Unlike existing integrative analyses that treat gene expression and TF-DNA binding as two separate processes, our approach puts these two types of data into one single model, which is more biologically meaningful. This statement does not explain well enough how the integration takes place. To me it was necessary to read this, the first result section and the methods to get enough understanding

#Response: We thank the reviewer for pointing this out. In order to make this point clear, we have revised the Introduction part, “In this study, we present a comprehensive methodology, LogicTRN, by integrating gene expression data and TF-DNA binding information to decipher TF regulatory logics in gene transcription. The newly developed method can quantitatively characterize logic relations between TFs by combining *cis*-regulatory logics and transcriptional kinetics in one single model framework” (Page 4 in the main manuscript). We believe this description can provide an idea about the LogicTRN. The details of how the integration work can be found in the Methods part.

p16 our method established a biological model which treats gene expression and TF-DNA binding as indivisible aspects of the gene transcription.

Repetition

#Response: We have removed this statement in the revised version.

p17 in agreement with many conclusions previously drawn from the biologic experiments. The

information from both computational and literature search "many conclusions previously drawn" and "both computational and literature search" are not clear

#Response: We have revised this statement/section in the discussion as follows: "...as well as their related pathways or functional processes, in agreement with viewpoints previously drawn from biological experiments. The result from both computational modelling and experimental consistency suggests that LogicTRN is suitable for decoding co-regulatory features of TFs and their target genes in other biological systems" (see page 15 in the main manuscript).

p17 Our method can be extended to the examination of gene expression regulated not only by TFs, but also miRNA, epigenetic regulators, or other factors. It can also be applied to dissect the roles of DNA or RNA binding proteins whose functions are not determined yet, or unknown cis-regulatory elements. What of this is special to your methods?

#Response: Thanks again for the comments. In response, we have revised this statement as follows: "In the future, it should be possible to extend our method to cover gene regulation by other regulators, such as epigenetic regulators, and other DNA binding proteins". (see page 16 in the main manuscript).

p17 Comparison with existing methods, our model not only exhibits detailed TF co-regulation patterns, What "TF co-regulation patterns" are you referring to?

#Response: We have already revised this statement in the Discussion as follows: "Through simulation and comparison studies, our model show its reliability and robustness in capturing TF regulatory logics and identifying targets of TFs, and in constructing TRNs" (see page 15 in the main manuscript).

Typos etc

p4 to develop a transcriptional regulatory model, LogicTRN, based which one is able to

#Response: We have revised this part as follows: "The newly developed method can quantitatively characterize logic relations between TFs by combining *cis*-regulatory logics and transcriptional kinetics in one single model framework. The derived TF logics are then used to infer the putative TF cooperation in regulating target genes so as to reconstruct TRNs" (see page 4 in the main manuscript).

p7 the same approaching on each gene

#Response: We have revised this part as follows: "Repeatedly applying the same procedure for each gene, we thus can reconstruct a TRN that links all the TF logics and their regulated genes" (see page 6 in the main manuscript).

p15 which makes it difficult for developing

#Response: We have revised this as follows: “One obstacle is that transcriptional kinetic function is essentially nonlinear, which makes it difficult to develop computational methodologies” (see page 14 in the main manuscript).

Reviewers' comments:

Reviewer #1 (Remarks to the Author):

The authors have very convincingly answered the comments and questions made in the initial review. I believe that the additional analysis strongly improve the manuscript. I would therefore strongly advise to include the figures 3 in the response letter as an additional supplementary figure in the manuscript, and comment the analysis on the negative controls.

With this modification (which the authors can be trusted to implement), I can recommend the manuscript for publication.

Reviewer #2 (Remarks to the Author):

I would like to thank the authors for the many improvements that make the paper easier to read and understand. However, there are still some important points that need clarification. Please provide binding data used for assessment and obtained as output from methods as supplemental data

Of particular importance are the section "Comparison with other methods" and the first section of the supplement, as they provide the only systematic data supporting the performance of inference. They need substantial improvement, in particular the supplemental section on Dataset One which is hard to read and understand. This is very important as simulated data may provide a good assessment on the inference capability of your method for the simulated framework, but not for real data.

* it is still not clear how assessment was performed based on Dataset One

- the binding data is both used as method input and for validation?

- please describe the set of positives and negatives, do they cover the entire genome, and at what resolution?

* human dataset

- TF binding motifs and previous methods constitute a bronze standard, but not a gold standard.

- Why not use ChIP seq data here as well as in the mouse case or use data where ChIP seq data is available?

- Otherwise, I would suggest to remove this human data from paper

* mouse dataset: this is the only rigid assessment based on a gold standard, but more analysis is necessary

- performance of logicTRN is so much better than of the other methods; this big improvement needs to be motivated more

- e.g., what is the intuition how all the methods use input data to assert binding, why is yours superior?

- also, even ChIP seq data (one of the best available datasets available so far) prone to false positive (e.g. spurious binding events without effect on expression) and false negatives (missing peaks): thus even almost perfect methods cannot achieve a perfect (AUC close to 1) score

- please also use AUPR for assessment, as AUC tends to inflate performance, as it focuses on negatives

Responses to reviewers' comments or concerns

Reviewer #1 (Remarks to the Author):

The authors have very convincingly answered the comments and questions made in the initial review. I believe that the additional analysis strongly improve the manuscript. I would therefore strongly advise to include the figures 3 in the response letter as an additional supplementary figure in the manuscript, and comment the analysis on the negative controls.

With this modification (which the authors can be trusted to implement), I can recommend the manuscript for publication.

#Response: We thank the reviewer for this very positive comment! As requested, we have added Figure 3 from our earlier response letter as an additional supplementary Figure S1. In the revised manuscript, we have also described analysis on the negative controls more clearly in the supplementary information Method Seven.

Reviewer #2 (Remarks to the Author):

I would like to thank the authors for the many improvements that make the paper easier to read and understand. However, there are still some important points that need clarification. Please provide binding data used for assessment and obtained as output from methods as supplemental data

#Response: According to the reviewer's suggestion, we have added TF binding data for mouse ESCs as supplemental Table S1. An additional Table S2 of known binding target genes of ESC TFs has been added, to help define positive binding data, also in response to your queries. This should clarify how we have defined negative versus positive binding data. The original supplemental Tables S1-6 also were renamed to Tables S3-8, both in the supplement and in the text.

Of particular importance are the section "Comparison with other methods" and the first section of the supplement, as they provide the only systematic data supporting the performance of inference. They need substantial improvement, in particular the supplemental section on Dataset One which is hard to read and understand. This is very important as simulated data may provide a good assessment on the inference capability of your method for the simulated framework, but not for real data.

#Response: As suggested, we describe how to assess the different methods in more detail in both the Results (Page 8) and Methods (Page 23) sections of main manuscript, and we provide new supplemental information. Specifically, we have addressed the questions related to methods comparisons in the revised manuscript. It includes 3 unique parts: 1) in the revised Dataset One supplemental information, it now only contains data source and ChIP-seq data processing; 2) Added "Comparison among different methods" in Methods section page 23. It describes how to perform analyses of ROC curve and Precision-Recall (PR) curve analyses for comparison study, and how to define negative and positive of TF target genes. 3) In the revised Results section page 8, it describes more clearly the results of both ROC and Precision-Recall (PR) analyses. This leads to the addition of a new Fig. 3B (replacing the previous one) that shows the PR curve and AUC scores of the various methods.

* it is still not clear how assessment was performed based on Dataset One

#Response: We utilized both ROC and Precision-Recall (PR) AUCs to evaluate different methods. In the revised manuscript, we have indicated how we performed the comparison studies based on the Dataset One (see page 8 in Results and page 23 in Methods sections of main manuscript). (see response above).

- the binding data is both used as method input and for validation?

#Response: In our previous comparison studies, we used input binding data for two sets of TFs: 1) ChIP-seq binding peaks of ESC-related regulators Oct4, Sox2, Nanog and Suz12, and 2) predicted binding motifs of Cancer-related TFs NFKB1, RELA, TP53 and TP63. The binding data were only used as method inputs but NOT as validation. For validation, we employed known binding target genes of TFs, for example NFKB1, RELA, TP53 and TP63 based on experimental testing of binding sites. We have built and reported the known target gene dataset of the 4 cancer TFs (Yan et al., 2013 Plos One, 8: e73656; Si et al., 2016 Oncogene, 35:5781-5794). Known target gene datasets of Oct4, Sox2, Nanog and Suz12 were built based on both ChIP-seq binding and experimental testing of gene expression by TF knockdown or overexpression (derived from our web server database of ChIP-Array2, Wang et al., Nucleic Acids Res 2015). We have provided these known target gene lists of Oct4, Sox2, Nanog and Suz12 in Table S2.

- please describe the set of positives and negatives, do they cover the entire genome, and at what resolution?

#Response: We thank the reviewer for this comment. Our response is detailed on page 23 in the Methods section of the main manuscript, and in Table S2. In essence, there are no gold standards to define positive or negative signals of TF target genes. In practice, the known TF targets, according to current experimental data or reports, are often considered as positive signals, while the undetected targets are treated as negative signals. The negative signal, however, does not firmly suggest that genes must not be regulated by TFs. Despite this limitation and inherent assumptions, to evaluate performance of TF target prediction in the submitted manuscript, we defined known TF targets as positive signals and unreported ones as negative ones.

Moreover, use of known TF target identification using different methods with this positive and negative definition should be, at least partially, effective in predicting novel TF target genes. In mouse ESC study, the TF target genes were derived from whole genome ChIP-seq binding on promoters, which should cover the entire genome. Therefore, our study has genome-wide resolution for both positive and negative binding. The ChIP-seq binding data and known target genes of TFs have been provided as Tables S1 and S2, respectively.

* human dataset

- TF binding motifs and previous methods constitute a bronze standard, but not a gold standard.

#Response: We agree with the reviewer's comment, but we do NOT wish to suggest that the binding motifs represent a gold standard. TF binding motifs are widely used to predict TF target genes and may be successful in some studies; however, the binding sites of TFs are based on computational prediction. They could generate false positives or false negatives. As suggested by the reviewer, we focused ChIP-seq binding peaks of mESC as binding data of TFs, and we have now removed human dataset that used TF binding motifs.

- Why not use ChiP seq data here as well as in the mouse case or use data where ChiP seq data is available? Otherwise, I would suggest to remove this human data from paper

#Response: As you suggested, we have removed the human datasets. As outlined in the manuscript, LogicTRN requires two input datasets: TF binding signals and time series gene expression data. Gene expression in a specific cell type should be consistent with ChIP-seq experiments if ChIP-seq TF binding data is used. Also, to assess the accuracy or efficiency of TF target gene prediction among different methods, we need known target genes of TFs for comparison standards. Cancer-related TFs NFKB1, RELA, TP53 and TP63 are widely studied and there are existing databases. We have therefore set up and reported the known target gene dataset of these 4 TFs (Yan et al., 2013 Plos One, 8: e73656; Si et al., 2016 Oncogene, 35:5781-5794). Because of these considerations and limited availability of cell-specific ChIP-seq data, we used the binding motif of these 4 TFs and cancer gene expression data in our previous submission; but based on your suggestion, we have removed the human dataset and its result in Results section of the revised manuscript and supplemental information.

* mouse dataset: this is the only rigid assessment based on a gold standard, but more analysis is necessary.

#Response: You are absolutely correct and we are grateful for the suggestion to perform additional analyses. In this revision, we carried out both ROC and Precision-Recall (PR) AUCs to assess performance of different models based on mouse datasets. In revised manuscript, the evaluation using both methods demonstrates that LogicTRN performs better than other methods. With simulation and application studies, we provide evidence that our method is effective and that it captures the nature of transcriptional gene regulation.

- performance of logicTRN is so much better than of the other methods; this big improvement needs to be motivated more

- e.g., what is the intuition how all the methods use input data to assert binding, why is yours superior?

#Response: To address your query, we respond to this comment with 3 major points:

1) LogicTRN is able to describe various modes of TF-TF interactions in cooperatively regulating target genes. Other existing methods, including COGRIM, APG, NCA and PTHGRN, mainly infer the regulatory relationships between TFs and genes, and are generally based on assumption that TFs independently contribute to gene regulation. LogicTRN uses logic operations to represent various modes of TF interactions. Therefore, we submit that LogicTRN can efficiently capture regulatory features of TFs, and provide extra capacity in accommodating the complex behaviors of TF interactions.

2) LogicTRN is capable of describing nonlinear gene regulation of TFs. It is well known that gene transcription tends to be nonlinear when TF concentrations are dynamic. For convenience of computation, existing methods actually use linear regression function to describe the TF contributions to regulate gene expression. Our model, however, is developed from nonlinear kinetic function of gene transcription, and can thus accommodate nonlinear behavior of TF gene regulation.

3) LogicTRN incorporates a variety of biological data in a single model framework. We utilize the concept of TF occupancy to bridge various biological quantities including TF gene expression, TF binding affinity and sites, and gene expression. On the one hand, TF occupancy characterizes the dynamic status of TF-DNA binding, which is determined by the TF concentration and TF binding affinity and sites on the promoter of the target genes. On the other hand, TF occupancy can be used to model the level of target gene expression. Moreover, we use logic gates to represent TF interactions.

Therefore, we characterize the regulatory complexity of TF interactions using such an integrative model. Solving the model equation can lead to identification of TF regulatory logics. To our knowledge, LogicTRN is the first method to computationally identify TF regulatory logics.

Finally in response to your point, the results we provide in this paper that involve comparisons, simulation and application studies demonstrate that the performance and robustness of LogicTRN is in many ways superior to other methods. We believe it provides a better methodology for dissecting dynamics and complexity of transcriptional gene regulation in biological systems.

- also, even ChIP seq data (one of the best available datasets available so far) prone to false positive (e.g. spurious binding events without effect on expression) and false negatives (missing peaks): thus even almost perfect methods cannot achieve a perfect (AUC close to 1) score

#Response: You are correct. It is nearly impossible to obtain an AUC of 1; however, data approaching an AUC of 1 should be indicative of an optimized method. ChIP-seq and binding motif data only represent possible physical sites of TF-DNA interactions. Identified binding sites could be false positives or false negatives, and are not necessary to have biological function, even for conserved binding motifs. As a result of our previous comparison study, we obtained an AUC score of 0.98 using predicted TF binding motif data. This is close to 1, but it is based on predicted datasets. Comparatively, experimentally derived ChIP-seq data achieved an ROC-AUC of 0.930. This value is less than that derived from predicted datasets, but it represents a reasonable assessment of experimental data. These analyses suggest that our model is highly predictive for experimental data, but one where false positives and negatives would still be expected. This is a limitation of the method addressed by ROC curve and AUPR (see below).

- please also use AUPR for assessment, as AUC tends to inflate performance, as it focuses on negatives

#Response: Thanks for the good suggestion. In response, we have re-evaluated the performance of the methods using AUPR (see page 8 in Results and page 23 in Methods sections of main manuscript). In revised manuscript, the evaluation using both ROC-AUC (0.930) and Precision-Recall (PR) AUC (0.933) methods demonstrates that LogicTRN can perform better than other methods. With simulation and application studies, we have provided evidence that our method is more effective and capture nature of transcriptional gene regulation.

Reviewers' comments:

Reviewer #2 (Remarks to the Author):

Dear authors, thank you very much again for the paper improvements and thoughtful answers to my comments. I must admit, I am still a little confused about the exact setup. As for this discussion, let's define the assessment standards for the set of positives and negatives you used for the calculation of AUPRs and AUCs (ignoring their experimental shortcomings for now). In order to allow readers to recapitulate the AUPRs and AUCs, they need to know

- 1- the exact set of positives
- 2- the exact set of negatives
- 3- the target predictions including their decision values for all methods.

In this context, I still don't see what you used as positive assessment standard. I assume it's Table S2 ("known binding target genes"). You state, "An additional Table S2 of known binding target genes of ESC TFs has been added, to help define positive binding data". So is Table S2 exactly the positive binding data used for calculating the AUPRs and AUCs or not? And, what is the relationship between S1 and S2 and its significance on the paper? And, does the set of positives and negatives add up to the number of genes in the mouse genome? Please also provide all the predictions (point 3 above).

It's crucial to enable the reader to exactly replicate your AUC and AUPR assessment. The reason I insist on this is that I find the AUPR with 93% suspiciously high. For example, in case of the prediction of Pou5f1 targets, the assessment standard positives from S2 are 514 and 294 in case of Suz12. With an AUPR of >90% you essentially claim to miss only 10% of these targets and predict only 10% false positives against a background of 20000 genes (negatives). Given our discussion the current "gold" standards aren't perfect, in terms of *their* limited recall and precision (as compared to the unknown "true" network) In case of your own method but also the comparison methods, I therefore fail to understand how you could reach such a high performance. I suspect, this my failure is still due to my lack of understanding how the assessment standard is constructed exactly.

Responses to Reviewer 2

Reviewer #2 (Remarks to the Author):

Dear authors, thank you very much again for the paper improvements and thoughtful answers to my comments. I must admit, I am still a little confused about the exact setup. As for this discussion, lets define the assessment standards for the set of positives and negatives you used for the calculation of AUPRs and AUCs (ignoring their experimental shortcomings for now). In order to allow readers to recapitulate the AUPRs and AUCs, they need to know

1- the exact set of positives

2- the exact set of negatives

3- the target predictions including their decision values for all methods.

#Response: Thanks for the comment, and we apologize for the continued confusion. In this study and in all our previous submissions, the comparisons used two input datasets: (1) genome wide ChIP-seq binding data of TFs Pou5f1 (Oct4), Sox2, Nanog and Suz12 in mouse ESC (i.e. Supplemental Table S1); and (2) gene expression data using Affymetrix Mouse Expression 430A Array covering genome scale ~14,000 well-characterized mouse genes (so-called MOE430A). We first generated the benchmark set consisting of a “positive” and a “negative” set. It included 3204 genes overlapped between the gene expression dataset (MOE430A) and the ChIP-seq binding data (Table S1). The reason we chose the benchmark set is because all the 5 algorithms require both binding data and gene expression as inputs.

More specifically:

- 1- the exact set of positives is defined for each individual TF and the set corresponds to the overlapped genes between the known target of each TF (as provided in the revised Table S2-I) and the benchmark set. The positive set of each TF is in revised Table S2-II;
- 2- the exact set of negatives is also defined for each individual TF and corresponds to the genes present in the benchmark set but which are not included in the positive set as defined in 1- above. This information is also provided in revised Table S2-III;
- 3- the target predictions, including their decision values for all 5 methods (LogicTRN, APG, COGRIM, NCA, and PTHGRN), are provided by the revised Table S2-IV.

Table S2, provided in the last submission, represents an old version of ChIP-Array mouse library which has been updated. In the present manuscript, the results of AUPR and ROC analyses are based on the latest version of ChIP-Array data and validation experiments. Revised Table S2-I thus reflects this updated information.

In this context, I still don't see what you used as positive assessment standard. I assume its Table S2 (“known binding target genes”). You state, “An additional Table S2 of known binding target genes of ESC TFs has been added, to help define positive binding data”. So is Table S2 exactly the positive binding data used for calculating the AUPRs and AUCs or not?

#Response: The positive assessment standard is described above (see 1-). Table S2 in the last submission did not correspond to the exact positives used for calculating the AUPRs and AUCs. Only overlapped genes among Table S2, Table S1 and the MOE430A genome were used as positives in the previous submission. As just mentioned, we have updated the known targets of TFs in revised Table S2-I. Consequently, the positive and negative sets are updated and defined as indicated above (see the revised Table S2-II and III, respectively).

And, what is the relationship between S1 and S2 and its significance on the paper? And, does the set of positives and negatives add up to the number of genes in the mouse genome? Please also provide all the predictions (point 3 above).

#Response: Table S1 represents genome-wide genes containing binding loci of the 4 TFs based on binding peak analysis of a ChIP-seq data (i.e. GSE11724), and it was utilized as input TF binding data for the 5 algorithms. Revised Table S2-I represents the target genes containing binding loci of the 4 TFs from *all* published ChIP-seq data in mouse and *validated* by experimental approaches. According to our definition, the set of positives and negatives corresponded to 3204 genes (i.e. the benchmark set), which represent a genome-wide set of overlaps among ChIP-seq binding and mouse MOE430A genome. The predictions of the 5 methods are provided in the revised Table S2-IV.

Its crucial to enable the reader to exactly replicate your AUC and AUPR assessment. The reason I insist on this is that I find the AUPR with 93% suspiciously high. For example, in case of the prediction of Pou5f1 targets, the assessment standard positives from S2 are 514 and 294 in case of Suz12. With an AUPR of >90% you essentially claim to miss only 10% of these targets and predict only 10% false positives against a background of 20000 genes (negatives). Given our discussion the current "gold" standards aren't perfect, in terms of *their* limited recall and precision (as compared to the unknown "true" network) In case of your own method but also the comparison methods, I therefore fail to understand how you could reach such a high performance. I suspect, this my failure is still due to my lack of understanding how the assessment standard is constructed exactly.

#Response: Thanks for your excellent comments and for pointing out this area of confusion. We agree the AUPR value looks high. To examine whether this result was from a biased comparison, we conducted following three tests (see below Figures 1-3).

In the previous submission, the positive and negative sets are highly imbalanced, i.e. the majority of genes in the benchmark set were negatives. To avoid the potential effects of such an imbalance on the ROC and PR curves, we adopted a "balanced" approach. According to Fawcell's suggestion¹, we selected the same number of negatives and positives to calculate AUC and AUPR. This can be carried out by randomly picking up the same number of positives from negative genes.

Firstly, we re-calculated the ROC and PR curves according to the updated Table S2 using the "balanced" approach, i.e. 100 random picks of the same number of positives as negatives. Noticed is that both curves and AUC/AUPR values are averages of the 4 TFs. As shown in Figure 1AB, we found that AUCs are similar to those provided in the last submission for each algorithm, and the AUPRs are comparable to the values in the last submission. Although the AUPR was somewhat higher in the previous submission, the difference observed here could be due to the adjustment of the positive and negative set in the updated Table S2. Regardless, these results support the superior performance of LogicTRN in TF target gene prediction relative to the other algorithms.

Figure 1. Curves and AUCs of ROC (A) and AUPR (B) for the 5 different algorithms using the same number of positives as negatives

Secondly, we performed the ROC and PR analyses using all of the positive and negative genes. The results are shown in Figure 2AB. Similar to the “balanced” approach, LogicTRN holds AUC of 0.935, which is higher than the other 4 algorithms. Compared with Figure 1B, PR curve of LogicTRN shows a “shifted” distribution (according to Fawcett’s definition). Its AUPR is 0.301. Although this value is much lower than the AUPR of Figure 1B (i.e. 0.891), it remains the highest among the 5 algorithms. The data are as follows:

Figure 2. Curves and AUCs of ROC (A) and AUPR (B) for the 5 different algorithms using all negatives

Thirdly, to examine effects of the scale of negatives, we increased the degree of imbalance by changing the numbers of negative. The result in Figure 3AB shows a corresponding trend in AUC and AUPR when the number of negatives changes. In Figure 3, scale 0 corresponds to the “balanced” approach (i.e. used in Figure 1AB), while scale 1 corresponds to the full size of the negative set (i.e. used in Figure 2AB). It can be seen that AUCs of ROC are constant, while AUPRs decrease as the scale of negative set increases. Despite the changes in AUPR, LogicTRN always performs the best at all scales of negative set.

Figure 3. Change in AUCs of ROC (A) and AUPR (B) for the 5 different algorithms using the different number of negatives. Scale 0 in X-axis is corresponding to the “balanced” approach, and scale 1 is corresponding to the full size of the negative set

According to the results above, we can obtain a high AUPR. For example, let’s consider two decision points in Figure 4A. On decision point 1, the recall= $TP/(TP+FN)$, is 0.5, and the precision= $TP/(TP+FP)$, is around 0.28. From Table S2-II and III, the average positives and negatives of 4 TFs are 141 and 3063, respectively. Therefore, we obtain $TP=70.5$, $FN=70.5$, $FP=181$ and $TN=2882$, which is corresponding to $TPR=50\%$ and $FPR=5.9\%$. On the other hand, for “balanced” approach in Figure 4B, the decision point 1 has recall of 0.5 and precision of around 0.9. Since both the positives and negatives are 141, we get $TP=70.5$, $FN=70.5$, $FP=8$ and $TN=133$, which is corresponding to $TPR=50\%$, and $FPR=5.7\%$. Therefore, we can see that TPRs and FPRs are consistent at decision point 1 for both approaches. Moreover, decision point 2 represents a critical performance of LogicTRN. In Figure 4A, recall is 0.95 and precision is around 0.26. Therefore, $TP=134$, $FN=7$, $FP=381$ and $TN=2682$, corresponding to $TPR=95\%$ and $FPR=12.4\%$, indicating a high prediction performance. Similar results can be achieved using the “balanced” approach.

In addition, as discussed above, only overlaps between Table S1 and mouse genome (MOE430A) are used as the benchmark set (3204 genes in last and current version). Therefore, background genes were actually not 20000.

Figure 4 Examples of two decision points in PR curves. Full scale of negatives (A) and “balanced” approach (B)

We agree that there are no “gold” standards to evaluate bioinformatics methods for TF target prediction. In general, it is difficult to collect complete sets of positive and negative signals. ChIP-seq binding data of TFs are widely used for ROC or AUPR analysis and are used to define positive and negative sets based on ChIP-seq binding signals^{2, 3, 4, 5}. Our defined sets were assumed to be biologically meaningful; however, they were conditional positives and negatives. We understand this standard has its limitations, however, the results in Figures 1-3 still demonstrate the advantage of LogicTRN over other algorithms, which is a major finding of our study.

Based on these revisions, several figures have been modified to show unbiased evaluations of target gene predictions based on the updated positive and negative datasets. We have replaced *former* Figures 3AB with Figures 2AB in the revised manuscript. *Revised* Figures 3AB now show analyses of ROC and AUPR using the updated set of positives and negatives in revised Table S2. The updated figures in the revised manuscript illustrate the robustness of LogicTRN for the reader. In summary, all three evaluation approaches utilized in the revised manuscript demonstrate that our LogicTRN is better than the other 4 algorithms in computational robustness.

Finally and to reiterate our response to one of your major comments, we describe the exact set of positives and negatives in “Comparison among different methods” page 23: “*First, we have generated a benchmark set consisting of a “positive” and a “negative” set. A set of genes in Table S2-I were used as positive signals if they are present in Affymetrix Mouse Expression 430A Array and also contain ChIP-seq binding loci of the four TFs based on binding peak analysis (see Table S1). The negative set includes those genes that are not positives but present in both Table S1 and the 430A Array (see Table S2-III). They represent genome-wide unknown binding target genes of the four TFs. The predicted target genes using the five methods were listed in Table S2-IV. To conduct an unbiased evaluation for ROC and PR analyses, we utilized full size of positives and negatives.*”, and corrected the Results section on page 8-9 by showing the new result of revised Figure 3AB.

References

1. Fawcett T. An introduction to ROC analysis. *Pattern Recognition Letters* **27**, 861-874 (2006).
2. Won KJ, Ren B, Wang W. Genome-wide prediction of transcription factor binding sites using an integrated model. *Genome biology* **11**, R7 (2010).
3. Siebert M, Soding J. Bayesian Markov models consistently outperform PWMs at predicting motifs in nucleotide sequences. *Nucleic acids research* **44**, 6055-6069 (2016).
4. Wang J, Lunnyak VV, Jordan IK. BroadPeak: a novel algorithm for identifying broad peaks in diffuse ChIP-seq datasets. *Bioinformatics* **29**, 492-493 (2013).
5. Huang F, Shen J, Guo Q, Shi Y. eRFSVM: a hybrid classifier to predict enhancers-integrating random forests with support vector machines. *Hereditas* **153**, 6 (2016).

REVIEWERS' COMMENTS:

Reviewer #2 (Remarks to the Author):

My thanks to the authors for their thoughtful response to my concerns and for the changes made to the paper. I have no further comments.